# PSformer: Parameter-efficient Transformer with Segment Shared Attention for Time Series Forecasting

## Abstract

Time series forecasting remains a critical challenge across various domains, often complicated by high-dimensional data and long-term dependencies. This paper presents a novel transformer architecture for time series forecasting, incorporating two innovative designs: parameter sharing module (PS) and Segment Shared Attention (SSA). The proposed model, PSformer, reduces the number of training parameters through the integrated parameter sharing mechanism without sacrificing performance. The spatiotemporal segment defined as a patch spanning across spatial variables and local time. The introduction of SSA could enhance the capability of capturing local spatio-temporal dependencies and improve global representation by integrating information across segments. Consequently, The combination of parameter sharing and SSA reduces the model's parameter count while enhancing forecasting performance. Extensive experiments on benchmark datasets demonstrate that PSformer outperforms many baseline approaches in terms of accuracy and scalability, positioning it as an effective and scalable tool for time series forecasting.

## 1 Introduction

Time series forecasting is an important learning task with significant application values in a wide range of domains, including the weather prediction Ren et al. (2021); Chen et al. (2023a), traffic flow Tedjopurnomo et al. (2020); Khan et al. (2023), energy consumption Liu et al. (2020); Nti et al. (2020), anomaly detection Zamanzadeh Darban et al. (2022) and the financial analysis Nazareth & Ramana Reddy (2023), etc. With the advancement of artificial intelligence techniques, significant efforts have been devoted to developing innovative models that continue to improve the prediction performance Liang et al. (2024); Wang et al. (2024b). In particular, the transformer-based model family has recently attracted more attention for its proved success in nature language processing OpenAI et al. (2024) and computer vision Liu et al. (2021); Dosovitskiy et al. (2021). Moreover, pre-trained large models based on the transformer architecture have shown advantages in time series forecastingLiu et al. (2024a); Jin et al. (2024); Chang et al. (2023); Woo et al. (2024), demonstrating that increasing the amount of parameters in transformer models and the volume of training data can effectively enhance the model capability.

On the other side, many linear models Zeng et al. (2023); Li et al. (2023) also show competitive performance compared to the complex designs of transformer-based models. One possible reason for their success in time series forecasting is that they have low model complexities and are less likely to overfit on noisy or irrelevant signals. As a result, even with limited data, these models can effectively capture robust information representations. To overcome the limitations of modeling long-term dependencies and capturing complex temporal relationships, PatchTST Nie et al. (2023) process temporal information by combining patching techniques to extract local semantic information, leading to superior performance. However, it applies channel-independent designs and leaves the significant potential for improvement in effectively modeling across variables. Moreover, the unique challenges of modeling multivariate time series data, where the temporal and spatial dimensions differ significantly from other data types, present many unexplored opportunities. While past research donghao & wang xue (2024); Zhang & Yan (2023); Ilbert et al. (2024) has largely treated these dimensions separately, the question of whether systematically mixing temporal and spatial information can further enhance model performance remains an open area for future investigation.

In this work, we explore innovative designs of the transformer-based model for time series forecasting by incorporating the spatiotemporal dependencies characteristic of time series tasks and a new parameter sharing model architecture. Past works proposed methods for capture spatiotemporal dependency information, such as MOIRAI Woo et al. (2024) flattens multivariate time series by treating all variables as a single sequence, and SAMformer Ilbert et al. (2024), which applies attention to the channel dimension to capture spatial dependencies. Unlike previous methods, we construct a transformer-based model called PSformer, which design as a two-stage SSA structure, each SSA consists of a parameter-sharing design. This parameter-sharing design (as PS Block) consists of three fully connected layers, which keeps the overall number of parameters in both the two-stage SSA and final fusion stage of the PSformer efficient, while also facilitating information sharing between different modules of the model. For SSA, we use patching to divide the variables into different patches, then identify the patches at the same position across different variables and merge them into a segment. As a result, each segment represents the spatial extension of a single-variable patch. In this way, we decompose the multivariate time series into multiple segments. Attention is applied within each segment to enhance the extraction of local spatial-temporal relationships, while information fusion across segments is performed to improve the overall predictive performance. Additionally, by incorporating the SAM optimization method, we further reduce over-fitting while maintaining efficient training. We conduct extensive experiments on long-term time series forecasting datasets to verify the effectiveness of the SSA structure. Our model demonstrates competitive performance in comparison to previous state-of-the-art models, achieving the best performance on 7 out of 8 mainstream long-term time series forecasting tasks. The contributions are summarized as follows:

- We developed a novel transformer-based model structure for time series forecasting, where the parameter sharing technique is applied in the transformer block to reduce the model complexity and improve the generalization ability.
- We proposed a SSA mechanism tailored for multi-variate data, which merges the temporal sequence of different channels to form a local segment and applies attention within each segment to capture both temporal dependencies and cross-channel interactions.
- Through extensive experiments in long-term forecasting tasks and ablation studies, we verified the effectiveness and superior performance of our proposed framework.

## 2 Related Work

### 2.1 Temporal Modeling in Time Series Forecasting

In recent years, time series analysis has received widespread attention, with more deep learning methods being applied to time series forecasting. These deep learning methods focus on establishing temporal dependencies within time series data to predict future trends. The models can be broadly categorized into RNN-based, CNN-based, MLP-based, and Transformer-based approaches. RNNs and their LSTM variants were widely used for time series tasks in the past, with related works such as DeepARSalinas et al. (2020). CNN-based methods like TCN Bai et al. (2018) and TimesNet Wu et al. (2023) have been designed to adapt convolutional structures specifically for temporal modeling. MLP-based approaches, such as N-BEATS Oreshkin et al. (2020), RLinear Li et al. (2023), and TSMixer Chen et al. (2023b), have demonstrated that even simple network structures can achieve solid predictive performance. Moreover, Transformer-based models have become increasingly popular in time series forecasting due to the unique attention mechanism of Transformers, which provides strong global modeling capabilities. Many recent works leverage this to enhance time series modeling performance, such as Informer Zhou et al. (2021), Autoformer Wu et al. (2021), Pyraformer Liu et al. (2022), and Fedformer Zhou et al. (2022). Additionally, PatchTST Nie et al. (2023) further divides time series data into different patches to enhance the ability to capture local information. However, the aforementioned models primarily focus on temporal modeling, with less emphasis on modeling the relationships between variables. Although PatchTST attempted to incorporate cross-channel designs, it observed degraded performance in their model.

On the other hand, some pre-trained large models have been applied to time series forecasting tasks Das et al. (2023); Liu et al. (2024a); Gao et al. (2024); Liu et al. (2024c); Zhou et al. (2023); Jin et al. (2024).

For example, MOMENT Goswami et al. (2024) uses the patching method and mask pre-training to build a pre-trained model for time series, while GPT4TS Zhou et al. (2023) also adopts the patching method and uses GPT2 as the backbone. The increase in model parameters has provided them with greater expressive power but also increased the difficulty of training.

## 2.2 Variate Modeling in Time Series Forecasting

In addition to modeling temporal dependencies, recent works have focused on modeling inter-variable dependencies donghao & wang xue (2024); Zhang & Yan (2023); Ilbert et al. (2024); Woo et al. (2024); Liu et al. (2024b). ModernTCN donghao & wang xue (2024) employs different 1-D convolutions to capture the temporal and variable dimensions separately; Crossformer Zhang & Yan (2023) processes temporal and spatial information separately via routing, followed by decoder-based fusion for prediction (denoted as S&M).; SAMformer Ilbert et al. (2024) focuses on channel-wise attention mechanisms but fails to incorporate temporal information interaction; MORAI Woo et al. (2024) "flattens" multivariate time series into univariate sequence tokens and utilizes a masking mechanism to train the foundation model; iTransformer Liu et al. (2024b) represents multivariate time series and captures global dependencies. All of these works emphasize the simultaneous modeling of both variable and temporal dependencies as critical directions for improving multivariate time series modeling, which helps establish global spatial-temporal dependencies. However, this may weaken the ability to capture local spatial-temporal dependencies. Additionally, expanding the global receptive field of spatio-temporal dependencies could increase model complexity, which in turn may lead to overfitting due to the larger number of parameters.

## 2.3 Parameter Sharing Structure

To reduce the model complexity in deep learning, parameter sharing is a crucial technique that can significantly reduce the amount model parameters and enhance computational efficiency. In CNNs, convolutional filters share weights across spatial locations, capturing local features with fewer parameters. Similarly, LSTM networks share weight matrices across time steps to manage memory and control information flow. By studying the sharing

Table 1: Architectural Comparison. S&M applies separate attention to temporal and spatial dimensions with fusion, while Joint employs simultaneous attention across both dimensions.

| Feature | Enc-In | X-Enc | Att-In | X-Att | ST |
|---|---|---|---|---|---|
| **Crossformer** | ✗ | ✗ | ✗ | ✗ | S&M |
| **Reformer** | ✗ | ✗ | ✔ | ✗ | — |
| **ALBERT** | ✗ | ✔ | ✗ | ✗ | — |
| **PSformer** | ✔ | ✗ | ✔ | ✔ | Joint |

of attention weights, Xiao et al. (2019) improves Transformer inference speed via parameter sharing mechanisms. Reformer Kitaev et al. (2020) employs query-key weight sharing in attention to cut costs while preserving accuracy (as Att-In). ALBERT Lan et al. (2020) extends parameter sharing across Encoder layers (as X-Enc) in natural language processing, reducing parameter redundancy while maintaining performance. In multi-task learning, the Task Adaptive Parameter Sharing approach Wallingford et al. (2022) selectively fine-tunes task-specific layers while maximizing parameter sharing across tasks, achieving efficient learning with minimal task-specific modifications. Those studies demonstrate that parameter sharing has the potential for model size reduction, generalization ability enhancement and mitigating the over-fitting risks across various tasks. Unlike previous methods, our method employs PSBlock, which combines intra-attention sharing and cross-attention sharing (as X-Att), forming the complete parameter architecture of PSEncoder layers (as Enc-In). The architectural differences among models regarding parameter sharing and spatio-temporal attention are summarized in Table 1.

# 3 The PSformer Framework

## 3.1 Problem Formulation

As shown in Figure 1, the input multivariate time series is denoted as $X \in \mathbb{R}^{L \times M}$ as with look-back window $L : (x_1, x_2, ..., x_L)$ and $M$ variables, where $x_t$ represents the $M$-dimensional vector at time step $t$. $L$ will be

equally divided into $N$ non-overlapping patches of size $P$. $P^{(i)}$ denote the $i$-th patch with size $P$, where $i = 1, 2, 3, ..., N$. The $P^{(i)}$ of the $M$ variables forms the $i$-th segment, which denote cross-channel patch of length $C$, where $C = M \times P$. The input time series X transformed from $X \in \mathbb{R}^{L \times M}$ to $X \in \mathbb{R}^{C \times N}$ by segment and time series transform (STF). The predict target is the future values of next $F$ time steps, e.g., $(x_{L+1}, ..., x_{L+F})$. Beside, we denote $\boldsymbol{X}^{in}$ and $\boldsymbol{X}^{out}$ as the input and output signals for the specified layers.

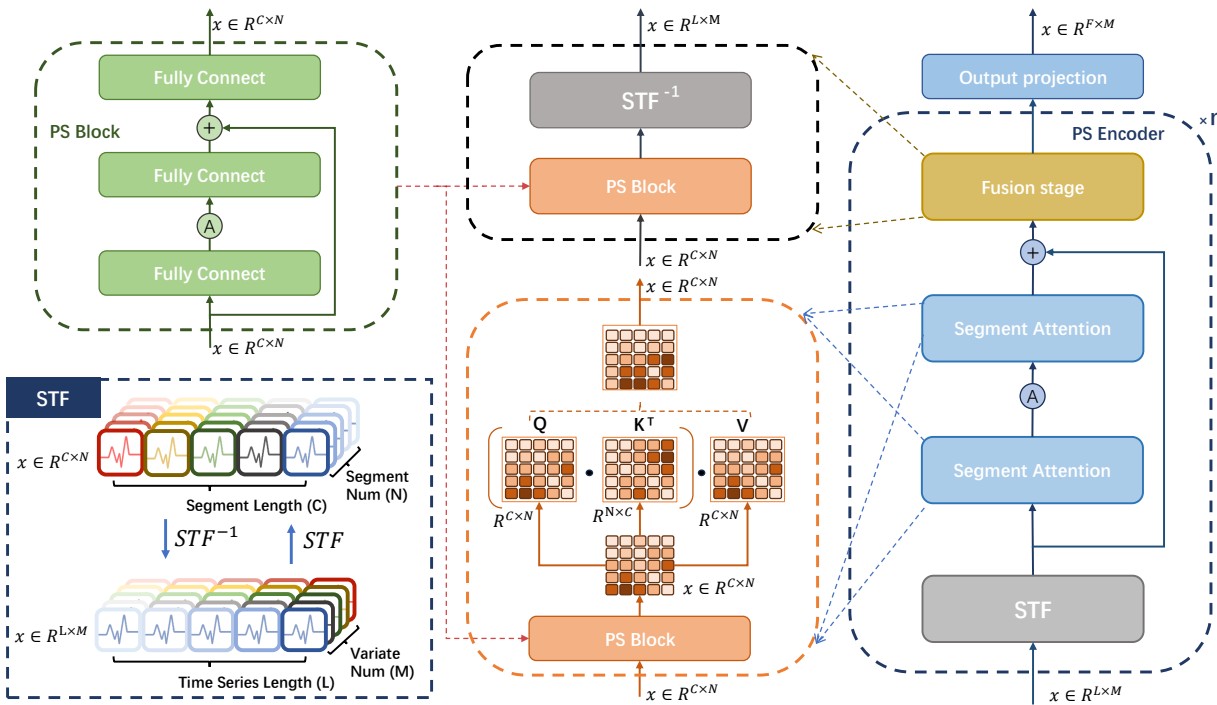

Figure 1: PSformer Network Structure. The PSformer model structure consists of the PS Block, which is composed of three fully connected layers that constitute all network parameters. These parameters are shared across the SSA and fusion stage in the PS Encoder. Through the STF process, SSA constructs an attention matrix along the segment dimension. In the two-stage segment shared attention, the PS Block generates matrices that are simultaneously used as the Q, K, and V matrices for the segment shared attention mechanism. Additionally, in the fusion stage, it further integrates feature information. The parameters of the PS Block form the entirety of the PS Encoder's parameters, which are finally transformed by the output projection for time-series prediction.

## 3.2 Model Structure

The constructed PSformer model is depicted in Figure 1, where the PSformer Encoder serves as the backbone of the model and the key components of the Encoder include the SSA and PS Block. The PS Block provides the parameters for all layers within the Encoder by utilizing the parameter sharing technique.

**Forward Process** The univariate time series of length $L$ for the $i$-th variable, starting at time index 1, is denoted as $x_1^{(i)} = (x_1^{(i)}, ..., x_L^{(i)})$, where $i = 1, ..., M$. Then the input $(x_1, ..., x_L)$ with $M$ dimensions is presented as $x_1 \in \mathbb{R}^{L \times M}$, and $x_1$ is used as the input to the transformer network structure. Similar to other time series forecasting methods, we use RevIN Kim et al. (2022), which is added at both the input and output of the model.

**Segment Shared Attention** We introduce segment shared attention (SSA), which aggregates patches from different channels at a local time period into a segment and establishes spatial-temporal relationships across different segments. While the network used to construct the Query, Key, and Value matrices in SSA is implemented by a network with shared parameters and nonlinear activation (PS Block). Specifically, the input time series $X \in \mathbb{R}^{L \times M}$ is first divided into $N$ non-overlapping components along the $L$ dimension,

where $L = P \times N$, and transformed into $X \in \mathbb{R}^{(M \times P) \times N}$. By concatenating the $M$ and $P$ dimensions, the input transform to $X \in \mathbb{R}^{C \times N}$, where $C = M \times P$, facilitating the subsequent cross-channel information fusion.

In this transformed space, identical $\boldsymbol{Q} \in \mathbb{R}^{C \times N}$, $\boldsymbol{K} \in \mathbb{R}^{C \times N}$, and $\boldsymbol{V} \in \mathbb{R}^{C \times N}$ matrices are generated by applying a shared block's non-linear projection (applied by PS Block), with weights $\boldsymbol{W} \in \mathbb{R}^{N \times N}$ used to project the input $X^{in} \in \mathbb{R}^{C \times N}$. The $\boldsymbol{Q}$ and $\boldsymbol{K}$ matrices are then multiplied using a dot-product operation to form the attention matrix $\boldsymbol{Q}K^T \in \mathbb{R}^{C \times C}$, which captures relationships between different spatial-temporal segments (in the $C$ dimension) and is used to act on the $\boldsymbol{V}$ matrix. While the computation of $\boldsymbol{Q}$, $\boldsymbol{K}$ and $\boldsymbol{V}$ involves non-linear transformations of the input $X^{in}$ across segments in the $N$ dimension, the scaled dot-product attention primarily applies attention across the $C$ dimension, allowing the model to focus on dependencies between spatial-temporal segments across channels and time.

This mechanism ensures that information from different segments is integrated through the computation of $Q$, $K$, and $V$. It also captures local spatial-temporal dependencies within individual segments by applying attention to the internal structure of each segment. Additionally, it captures long-term dependencies across segments over larger time steps. The final output is $X^{out} \in \mathbb{R}^{C \times N}$, completing the attention process.

**Parameter Shared Block** In our work, we propose a novel Parameter Shared Block (PS Block), which consists of three fully connected layers with residual connections, as illustrated in Figure 1. Specifically, we construct three trainable linear mappings $\boldsymbol{W}^{(j)} \in \mathbb{R}^{N \times N}$ with $j \in \{1, 2, 3\}$. The output of the first two layers is computed as:

$$\boldsymbol{X}^{out} = \text{GeLU}(\boldsymbol{X}^{in}\boldsymbol{W}^{(1)})\boldsymbol{W}^{(2)} + \boldsymbol{X}^{in}, \tag{1}$$

which follows a similar structure as the feed-forward network (FFN) with residual connections. This intermediate output $\boldsymbol{X}^{out}$ is then used as the input for the third transformation, yielding

$$\boldsymbol{X}^{out} = \boldsymbol{X}^{in}\boldsymbol{W}^{(3)}. \tag{2}$$

Therefore, the PS Block as a whole can be expressed as:

$$\boldsymbol{X}^{out} = (\text{GeLU}(\boldsymbol{X}^{in}\boldsymbol{W}^{(1)})\boldsymbol{W}^{(2)} + \boldsymbol{X}^{in})\boldsymbol{W}^{(3)}. \tag{3}$$

And we denote PS Block output as $\boldsymbol{X}^{out} = \boldsymbol{X}^{in}\boldsymbol{W}^S$, where $W^S \in \mathbb{R}^{N \times N}$ and $\boldsymbol{X}^{out} \in \mathbb{R}^{C \times N}$. The structure of the PS Block allows it to perform nonlinear transformations while preserving a linear transformation path. Although the three layers within the PS Block have different parameters, the entire PS Block is reused across different positions in the PSformer Encoder, ensuring that the same block parameters $\boldsymbol{W}^S$ are shared across those positions, as illustrated in Figure 1. Specifically, PS Block share parameters across three parts of each PSformer Encoder, which includes two SSA layers and the final PS Block. In the SSA layer, the PS block outputs are used as the $Q$, $K$, and $V$ matrices to construct the attention mechanism. This parameter-sharing strategy reduces the overall number of parameters while maintaining the network's expression capacity.

**PSformer Encoder** In the PSformer Encoder, as shown in Figure 1, each layer shares the same parameters $\boldsymbol{W}^S$ of PS Block. For the input $\boldsymbol{X}^{in}$, the transformation in the PSformer Encoder can be expressed as follows:

**SSA stage one** is represented as: $\boldsymbol{Q}^{(1)} = \boldsymbol{X}^{in}\boldsymbol{W}^S$, $\boldsymbol{K}^{(1)} = \boldsymbol{X}^{in}\boldsymbol{W}^S$, $\boldsymbol{V}^{(1)} = \boldsymbol{X}^{in}\boldsymbol{W}^S$, Therefore, we have $\boldsymbol{Q}^{(1)}$, $\boldsymbol{K}^{(1)}$, $\boldsymbol{V}^{(1)} \in \mathbb{R}^{C \times N}$. The dot-product attention operation can be formulated by

$$\boldsymbol{O}^{(1)} = \text{Softmax}(\frac{\boldsymbol{Q}^{(1)}\boldsymbol{K}^{(1)T}}{\sqrt{d_k}})V^{(1)}, \tag{4}$$

which is followed by the ReLU activation:$\boldsymbol{O}^{(1)}_{act} = ReLU(\boldsymbol{O}^{(1)})$.

**SSA stage two** is represented as: $\boldsymbol{Q}^{(2)} = \boldsymbol{O}^{(1)}_{act}\boldsymbol{W}^S$, $\boldsymbol{K}^{(2)} = \boldsymbol{O}^{(1)}_{act}\boldsymbol{W}^S$, $\boldsymbol{V}^{(2)} = \boldsymbol{O}^{(1)}_{act}\boldsymbol{W}^S$. Similarly, we have $\boldsymbol{Q}^{(2)}$, $\boldsymbol{K}^{(2)}$, $\boldsymbol{V}^{(2)} \in \mathbb{R}^{C \times N}$, and the dot-product attention operation

$$\boldsymbol{O}^{(2)} = \text{Softmax}(\frac{\boldsymbol{Q}^{(2)}\boldsymbol{K}^{(2)T}}{\sqrt{d_k}})\boldsymbol{V}^{(2)}. \tag{5}$$

Table 2: Datasets for long-term forecasting. Dataset size is structured as (Train, Validation, Test).

| Dataset | Variate | Predict Length | Frequency | Dataset Size | Information |
|---|---|---|---|---|---|
| ETTh1,ETTh2 | 7 | {96,192,336,720} | Hourly | (8545, 2881, 2881) | Electricity |
| ETTm1,ETTm2 | 7 | {96,192,336,720} | 10 mins | (34465, 11521, 11521) | Electricity |
| Weather | 21 | {96,192,336,720} | 15 mins | (36792, 5271, 10540) | Weather |
| Electricity | 321 | {96,192,336,720} | Hourly | (18317, 2633, 5261) | Electricity |
| Exchange | 8 | {96,192,336,720} | Daily | (5120, 665, 1422) | Exchange rate |
| Traffic | 862 | {96,192,336,720} | Hourly | (12185, 1757, 3509) | Transportation |

Table 3: Long-term forecasting task. All the results are averaged from 4 different prediction lengths {96, 192, 336, 720}. A lower MSE or MAE indicates a better performance. See Table B.2 in Appendix for the full results with more baselines.

| Metric | PSformer | | SAMformer | | TSMixer | | PatchTST | | MOMENT | | ModernTCN | | FEDformer | | GPT4TS | | Autoformer | | RLinear | | iTransformer | |
|---|---|---|---|---|---|---|---|---|---|---|---|---|---|---|---|---|---|---|---|---|---|---|
| | MSE | MAE | MSE | MAE | MSE | MAE | MSE | MAE | MSE | MAE | MSE | MAE | MSE | MAE | MSE | MAE | MSE | MAE | MSE | MAE | MSE | MAE |
| ETTh1 | 0.397 | 0.418 | 0.410 | 0.424 | 0.420 | 0.431 | 0.468 | 0.455 | 0.418 | 0.436 | 0.421 | 0.432 | 0.428 | 0.454 | 0.428 | 0.426 | 0.473 | 0.477 | 0.446 | 0.434 | 0.454 | 0.448 |
| ETTh2 | 0.338 | 0.390 | 0.344 | 0.391 | 0.354 | 0.400 | 0.387 | 0.407 | 0.352 | 0.394 | 0.343 | 0.393 | 0.388 | 0.434 | 0.355 | 0.395 | 0.422 | 0.443 | 0.374 | 0.398 | 0.383 | 0.407 |
| ETTm1 | 0.342 | 0.372 | 0.373 | 0.388 | 0.378 | 0.392 | 0.387 | 0.400 | 0.344 | 0.379 | 0.361 | 0.384 | 0.382 | 0.422 | 0.351 | 0.383 | 0.515 | 0.493 | 0.414 | 0.407 | 0.407 | 0.410 |
| ETTm2 | 0.251 | 0.313 | 0.269 | 0.327 | 0.283 | 0.339 | 0.281 | 0.326 | 0.259 | 0.318 | 0.262 | 0.322 | 0.292 | 0.343 | 0.267 | 0.326 | 0.310 | 0.357 | 0.286 | 0.327 | 0.288 | 0.332 |
| Weather | 0.225 | 0.264 | 0.261 | 0.293 | 0.255 | 0.289 | 0.259 | 0.281 | 0.228 | 0.270 | 0.237 | 0.274 | 0.310 | 0.357 | 0.237 | 0.271 | 0.335 | 0.379 | 0.272 | 0.291 | 0.258 | 0.278 |
| Electricity | 0.162 | 0.255 | 0.181 | 0.275 | 0.198 | 0.296 | 0.205 | 0.290 | 0.165 | 0.260 | 0.160 | 0.255 | 0.207 | 0.321 | 0.167 | 0.263 | 0.214 | 0.327 | 0.219 | 0.298 | 0.178 | 0.270 |
| Exchange | 0.358 | 0.399 | 0.445 | 0.470 | 0.532 | 0.523 | 0.367 | 0.404 | 0.437 | 0.446 | 0.555 | 0.536 | 0.478 | 0.477 | 0.371 | 0.409 | 0.613 | 0.539 | 0.378 | 0.417 | 0.360 | 0.403 |
| Traffic | 0.400 | 0.274 | 0.425 | 0.297 | 0.439 | 0.315 | 0.481 | 0.304 | 0.415 | 0.293 | 0.414 | 0.283 | 0.604 | 0.372 | 0.414 | 0.295 | 0.617 | 0.384 | 0.626 | 0.378 | 0.428 | 0.282 |
| Count | 7 | 8 | 0 | 0 | 0 | 0 | 1 | 0 | 0 | 0 | 1 | 1 | 0 | 0 | 0 | 0 | 0 | 0 | 0 | 0 | 1 | 1 |

The two-stage SSA mechanism can be viewed as analogous to an FFN layer, where the MLP is replaced with attention operations. Additionally, residual connections are introduced between the input and output, and the result is then fed into the final PS Block. Since the transformation in the final PS Block is represented as $\boldsymbol{O}^{out} = \boldsymbol{O}^{in}\boldsymbol{W}^S$, the entire encoder can be expressed as

$$\boldsymbol{X}^{out} = (\text{Attention}^{(2)}(\text{ReLU}(\text{Attention}^{(1)}(\boldsymbol{X}^{in}))) + \boldsymbol{X}^{in})\boldsymbol{W}^S, \tag{6}$$

with $\boldsymbol{X}^{out} \in \mathbb{R}^{C \times N}$. Finally, since $C = M \times P$ and $L = P \times N$, we apply a dimensionality transformation to obtain $\boldsymbol{X}^{out} \in \mathbb{R}^{L \times M}$.

After passing through $n$ layers of the PSformer Encoder, the final output is $\boldsymbol{X}^{pred} = \boldsymbol{W}^F\boldsymbol{X}^{out}$, where $\boldsymbol{X}^{pred} \in \mathbb{R}^{F \times M}$, $\boldsymbol{W}^F \in \mathbb{R}^{F \times L}$ is a linear mapping and $F$ is the prediction length. The $\boldsymbol{X}^{pred}$ is the final output of the PSformer model. The SSA mixes local spatio-temporal information without the need for positional encoding, and we provide a more detailed discussion in Appendix A.6.

# 4 Experiment

**Datasets** In this paper, we focus on the long-term time series forecasting. We follow the time series forecasting work in Ilbert et al. (2024) and use 8 mainstream datasets to evaluate the performance of our proposed PSformer model. As shown in Table 2, these datasets include 4 ETT datasets (ETTh1, ETTh2, ETTm1, ETTm2), as well as Weather, Traffic, Electricity, and Exchange. These datasets have been used as benchmark evaluations in many previous time series forecasting studies.

**Baselines** We select state-of-the-art (SOTA) models in the field of long-term time series forecasting, including not only Transformer-based models but also large models and other SOTA models. Specifically, baselines include (1) Transformer-based model: SAMformer Ilbert et al. (2024), iTransformer Liu et al. (2024b), PatchTST Nie et al. (2023), FEDformer Zhou et al. (2022), Autoformer Wu et al. (2021), (2) Pretrained Large

model: MOMENT Goswami et al. (2024), GPT4TS Zhou et al. (2023), (3) TCN-based model: ModernTCN donghao & wang xue (2024) and (4) MLP-based methods: TSMixer Chen et al. (2023b), RLinear Li et al. (2023). Additionally, we provide more baselines for a comprehensive comparison, including TimeMixer Wang et al. (2024a), CrossGNN Huang et al. (2023), MICN Wang et al. (2023), TimesNet Wu et al. (2023), FITS Xu et al. (2024), Crossformer Zhang & Yan (2023), PDF Dai et al. (2024), and TimeLLM Jin et al. (2024) Further details about these baselines can be found in Appendix B.8.2.

**Experimental Settings** The input time series length $T$ is set to 512, and four different prediction lengths $H \in \{96, 192, 336, 720\}$ are used. Evaluation metrics include Mean Squared Error (MSE) and Mean Absolute Error (MAE). We train our constructed models using the SAM optimization technique as in Ilbert et al. (2024). Setting the look-back window of RevIN to 16 for the Exchange dataset, more details about this setting can be found in Appendix B.7. We collect baseline results of SAMformer, TSMixer, FEDformer and Autoformer from Ilbert et al. (2024), in which SAMformer and TSMixer are also trained with SAM, and the results of iTransformer, RLinear and PatchTST from Liu et al. (2024b). We test ModernTCN with the fixed input length of 512 following donghao & wang xue (2024). For the pre-trained large models MOMENT and GPT4TS, the results are collected from Goswami et al. (2024). More details about the baseline settings can be found in Appendix A.2. Besides, We also provide results under uni-variate series and unrelated variate in the Appendix B.6.

## 4.1 Results and Analysis

**Summary of Experimental Results** The main experimental results are reported in Table 3. PSformer achieved the best performance on 7 out of 8 major datasets in long-term time series forecasting tasks, demonstrating its strong predictive capabilities across various time series prediction problems. This success is attributed to its SSA mechanism, which enhances the extraction of spatial-temporal information, and the parameter-sharing structure, which improves the model's robustness. The complete experimental results are in Appendix B.2. We also provide additional visualization results to analyze the attention maps across spatial and temporal dimensions in the Appendix B.5.

**Comparison with Other SOTA models** Compared to other Transformer-based models, PSformer demonstrates higher predictive accuracy, reflecting the effectiveness of dividing multivariate time series data into spatial-temporal segments for attention calculation. The neural network's ability to extract information from all spatial-temporal segments enhances the prediction performance. In contrast to current large pre-trained models, PSformer not only achieves better accuracy but also reduces the amount of parameters through parameter sharing. Although linear models are simpler and have fewer parameters with satisfactory performance in some cases, the ability to extract rich information from complex data is limited. In contrast, PSformer integrates the residual connections, thus enabling a linear path while retaining the capability to process complex nonlinear information. Moreover, the ConvFFN component in ModernTCN tailored for temporal data also confirms that the convolutional mechanism, which actually embodies the idea of parameter sharing, is also effective in the time series domain. With the same spirit, we have successfully applied the parameter sharing to the transformer-based models in the time series field and achieved superior performance.

Table 4: The MSE results of different number of segments ($N$) on the ETTh1 and ETTm1 dataset.

| # of Seg. | ETTh1 | | | | ETTm1 | | | | ETTh2 | | | |
|---|---|---|---|---|---|---|---|---|---|---|---|---|
| | **96** | **192** | **336** | **720** | **96** | **192** | **336** | **720** | **96** | **192** | **336** | **720** |
| 8 | 0.362 | 0.406 | 0.670 | 0.451 | 0.290 | 0.325 | 0.357 | 0.412 | 0.273 | 0.333 | 0.353 | 0.387 |
| 16 | 0.352 | 0.417 | 0.424 | 0.446 | 0.285 | 0.323 | 0.353 | 0.412 | 0.273 | 0.333 | 0.357 | 0.387 |
| 32 | 0.352 | 0.386 | 0.410 | 0.440 | 0.282 | 0.321 | 0.352 | 0.413 | 0.272 | 0.335 | 0.356 | 0.389 |
| 64 | 0.354 | 0.389 | 0.412 | 0.446 | 0.288 | 0.325 | 0.355 | 0.417 | 0.275 | 0.337 | 0.357 | 0.394 |

**Comparison with PatchTST and SAMformer** PatchTST employs a channel-independent design and divides time series data into multiple patches, which demonstrates the effectiveness of the patching method in time series processing. However, its channel-independent approach does not fully consider the relationships between different channels, focusing only on processing each channel individually. On the other hand, SAMformer applies attention directly to the channel dimension via dimension transformations and utilizes a simplified model structure, achieving good predictive performance. However, it may fail to capture valuable local information without the patching method. PSformer combines the advantages of both models while addressing their limitations. By using SSA, PSformer effectively captures local temporal information and handles relationships among different channels. This design enables PSformer to outperform them in various time series forecasting tasks as validated by extensive experiments.

## 4.2 Ablation Studies

**The Effect of Segments Numbers** Since PSformer employs a non-overlapping patching to construct segments, the model's performance may affected by the number of segments. Therefore, we tested the model's performance with different segment numbers on two datasets, ETTh1, ETTh2 and ETTm1. Given that the input sequence length is fixed at 512, the number of segments must be a divisor of the sequence length. Consequently, we set the number of segments to 8, 16, 32, and 64, and test the model on four different forecasting horizons. The test results are shown in Table 4, which indicate that the number of segments impacts the model's prediction accuracy to some extent, while the differences are relatively small. Across both datasets, a moderate number of segments (such as 32) tends to perform the balance for both short and long forecasting horizons.

**The Effect of PSformer Encoder Numbers**

Since PSformer adopts the SSA mechanism, with non-shared PS Block parameters across different Encoders (shared within Encoder), we tested the impact of varying the number of encoders on model performance. We conducted tests on the ETTh1 and ETTm1 datasets, varying the number of encoders from 1 to 4 with four different forecasting horizons. The experimental results are shown in Table 5. The results indicate that ETTm1 performs best with 3 encoders, while ETTh1 achieves better performance with just 1 encoder. This may suggest that for smaller datasets, fewer encoders result in better performance, as reducing the number of encoders decreases the amount of model parameters, thereby mitigating the risk of over-fitting.

Table 5: The MSE results of different number of encoders on the ETTh1 and ETTm1 dataset.

| # of Enc | ETTh1 | | | | ETTm1 | | | |
|---|---|---|---|---|---|---|---|---|
| | 96 | 192 | 336 | 720 | 96 | 192 | 336 | 720 |
| 1 | 0.352 | 0.385 | 0.411 | 0.440 | 0.288 | 0.324 | 0.356 | 0.414 |
| 2 | 0.355 | 0.392 | 0.418 | 0.443 | 0.284 | 0.323 | 0.356 | 0.415 |
| 3 | 0.355 | 0.391 | 0.416 | 0.443 | 0.282 | 0.321 | 0.352 | 0.413 |
| 4 | 0.355 | 0.389 | 0.416 | 0.440 | 0.282 | 0.321 | 0.353 | 0.416 |

**Ablation of Parameter Sharing Methods**

We investigate the impact of parameter-sharing mechanism on the model performance. In addition to the default parameter-sharing approach, which shares parameters only within encoder (`In-Encoder`), we also test the following approaches: a. no parameter sharing, i.e., `None`; b. sharing parameters only across encoders (with different parameters used for the PS blocks within each encoder), i.e., `Cross-Encoders`; and c. sharing parameters both within and across encoders, i.e., `ALL`. We conducted experiments on the ETTm1 and Weather datasets,

Table 6: Parameter Sharing with different methods

| Sharing Methods | | Cross-Enc | In-Enc | ALL | None |
|---|---|---|---|---|---|
| ETTm1 | 96 | 0.297 | 0.282 | 0.288 | 0.295 |
| | 196 | 0.329 | 0.321 | 0.326 | 0.338 |
| | 336 | 0.360 | 0.352 | 0.358 | 0.372 |
| | 720 | 0.420 | 0.413 | 0.414 | 0.425 |
| Weather | 96 | 0.158 | 0.149 | 0.151 | 0.154 |
| | 196 | 0.201 | 0.193 | 0.196 | 0.198 |
| | 336 | 0.252 | 0.245 | 0.245 | 0.245 |
| | 720 | 0.319 | 0.314 | 0.316 | 0.319 |

and the results are shown in Table 6. As can be seen, the `In-Encoder` method performs the best, followed by `ALL`, while `None` shows the worst performance. This indicates that the parameter-sharing mechanism contributes to improving the model performance. Furthermore, we provide a comparison of convergence rates between parameter sharing and non-parameter sharing in Appendix B.9, as well as a comparison of parameter savings achieved by parameter sharing in Appendix B.10 and Appendix A.4.

**Ablation of Segment Attention methods**

As an important component of PSformer, we designed SSA to effectively capture temporal information while fully utilizing the correlations between channels. In the ablation study, we compared different attention mechanisms without using SSA under parameter sharing, as well as the performance of using only MLP. We conducted experiments on the ETTh1 and ETTh2 datasets. In this study, w/o SSA means applying the attention mechanism to the input $x \in \mathbb{R}^{B \times M \times L}$ on the $M \times L$ dimensions. For the setting with CI (channel independence), it applies the attention mechanism on the $N \times P$ dimensions. The results in Table 7 show the performance of PSformer variants with different attention mechanisms on the ETTh1 and ETTh2 datasets across various forecasting horizons:

Table 7: Ablation analysis of SSA on ETTh1 and ETTh2 (MSE reported). Red: the best.

| Variant | ETTh1 | | | | ETTh2 | | | |
|---|---|---|---|---|---|---|---|---|
| | 96 | 192 | 336 | 720 | 96 | 192 | 336 | 720 |
| **w SSA** | 0.352 | 0.385 | 0.411 | 0.440 | 0.272 | 0.335 | 0.356 | 0.389 |
| **w/o SSA** | 0.369 | 0.397 | 0.414 | 0.448 | 0.288 | 0.365 | 0.373 | 0.398 |
| **w CI** | 0.376 | 0.407 | 0.427 | 0.455 | 0.285 | 0.382 | 0.369 | 0.395 |
| **only mlp** | 0.379 | 0.399 | 0.426 | 0.450 | 0.282 | 0.352 | 0.358 | 0.398 |

- The default PSformer configuration (with SSA) consistently achieves the lowest MSE across all horizons, demonstrating the effectiveness of SSA.

- When SSA is removed, the cross-channel attention is less effective at capturing both local and global temporal dependencies, resulting in slightly increased MSE .

- The variant with channel-independent attention shows further degradation, suggesting that neglecting inter-channel correlations impacts the model's ability to capture temporal features.

- Using only MLP layers also results in higher MSE. Although it performs second best for the three forecasting horizons on ETTh2, it still falls short compared to the performance with SSA, highlighting the necessity of applying SSA.

## 5 Conclusion and Future Work

In this work, we proposed the PSformer model for multivariate time series forecasting, which leverages SSA to facilitate information transfer among time series variables and capture spatial-temporal dependencies. By employing parameter sharing, the model effectively improves parameter efficiency and reduces the risk of overfitting when the data size is relatively limited. Overall, this designed network structure achieves state-of-the-art performance on long-term multivariate forecasting tasks by enhancing model parameter efficiency and improving the utilization of channel-wise information. Future work can focus on applying these techniques to the development of pre-trained large models for time series forecasting, to overcome the issue of excessively large parameter counts in existing pre-trained models, and improve the capability of extracting information from multivariate time series.

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

# A  Experimental Configuration

## A.1  Hardware

All experiments were conducted on two servers, each equipped with an 80GB NVIDIA A100 GPU and 4 Intel Xeon Gold 5218 CPUs.

## A.2  Details of Baseline Settings

We conducted all of our experiments with look-back window $L = 512$ and prediction horizons $H \in \{96, 192, 336, 720\}$. Results of PSformer and ModernTCN reported in Table 13 come from our own experiments. The difference between our experiment with ModernTCN and its official code is that we standardized the look-back window to 512 and set *drop_last=False* for the test set in the dataloader to ensure consistency with our experimental settings for a fair comparison. For MOMENT and GPT4TS, the results are collected from Goswami et al. (2024), except Exchange rate (which is tested on official code repository). The results of SAMformer, TSMixer, FEDformer and Autoformer are obtained from Ilbert et al. (2024), while the results of iTransformer, PatchTST, and RLinear are taken from Liu et al. (2024b).

## A.3  Settings for PSformer

We provides an overview of the experimental configurations for the PSFormer model across various tasks and datasets in Table 8. The experiments cover multiple time-series datasets, including ETTh1, ETTh2, ETTm1, ETTm2, Weather, Electricity, Traffic, and Exchange. In all experiments, the input sequence length (Input Length $T$) is set to 512, and each input is equally divided into 32 non-overlapping segments (Segments Num. $N$). The model architecture uses 3 Encoders for tasks, while for the ETTh1, ETTh2, ETTm2 and Exchange, 1 Encoder are used.

The learning rate adjustment strategy (schedule) is set to "constant" for all experiments, with a fixed learning rate (LR) of $10^{-4}$. The loss function used in all experiments is Mean Squared Error (MSE). The batch size is set to 16 for most tasks, except for the Traffic dataset, where the batch size is 8. Each experiment runs for 300 epochs, with a patience value of 30 for early stopping. A fixed random seed of 1 is applied across all experiments to ensure reproducibility.

## A.4  Model Size Comparison

Table 9 presents a comparison of the parameter size across different models, including the PSFormer and other baseline models such as SAMformer, TSMixer, ModernTCN, and RLinear. The comparison is conducted on ETTh1, Weather, and Traffic datasets, with prediction horizons $H \in \{96, 192, 336, 720\}$. PSFormer is evaluated in two configurations: the full model and the encoder part. The parameters of the encoder part refer to the number of parameters after excluding the linear mapping in the output layer. The table denote

Table 8: An overview of the experimental configurations for PSFORMER.

| Task-Dataset | Encoder Num. | Input Length $T$ | Segment Num. $N$ | schedule | LR* | Loss | Batch Size | Epochs | patient | random seed |
|---|---|---|---|---|---|---|---|---|---|---|
| ETTh1 | 1 | 512 | 32 | constant | $10^{-4}$ | MSE | 16 | 300 | 30 | 1 |
| ETTh2 | 1 | 512 | 32 | constant | $10^{-4}$ | MSE | 16 | 300 | 30 | 1 |
| ETTm1 | 3 | 512 | 32 | constant | $10^{-4}$ | MSE | 16 | 300 | 30 | 1 |
| ETTm2 | 1 | 512 | 32 | constant | $10^{-4}$ | MSE | 16 | 300 | 30 | 1 |
| Weather | 3 | 512 | 32 | constant | $10^{-4}$ | MSE | 16 | 300 | 30 | 1 |
| Electricity | 3 | 512 | 32 | constant | $10^{-4}$ | MSE | 16 | 300 | 30 | 1 |
| Traffic | 3 | 512 | 32 | constant | $10^{-4}$ | MSE | 8 | 300 | 30 | 1 |
| Exchange | 1 | 512 | 32 | constant | $10^{-4}$ | MSE | 16 | 300 | 30 | 1 |

Table 9: Comparison of the trainable model parameters

| Dataset | Horizon | PSformer | | SAMformer | TSMixer | ModernTCN | RLinear |
|---|---|---|---|---|---|---|---|
| | | *Full* | *Encoder* | | | | |
| ETTh1 | H=96 | 52,416 | 3,168 | 50,272 | 124,142 | 876,064 | 49248 |
| | H=192 | 101,664 | 3,168 | 99,520 | 173,390 | 1,662,592 | 98496 |
| | H=336 | 175,536 | 3,168 | 173,392 | 247,262 | 2,842,384 | 172368 |
| | H=720 | 372,528 | 3,168 | 369,904 | 444,254 | 5,988,496 | 369360 |
| Relative Size (Avg) | | 1.0 | 0.014 | 0.987 | 1.408 | 16.192 | 0.948 |
| Weather | H=96 | 58,752 | 9,504 | 50,272 | 121,908 | 2,709,280 | 49248 |
| | H=192 | 108,000 | 9,504 | 99,520 | 171,156 | 3,495,808 | 98496 |
| | H=336 | 181,872 | 9,504 | 173,392 | 245,028 | 4,675,600 | 172368 |
| | H=720 | 378,864 | 9,504 | 369,904 | 442,020 | 7,821,712 | 369360 |
| Relative Size (Avg) | | 1.0 | 0.039 | 0.953 | 1.347 | 25.708 | 0.948 |
| Traffic | H=96 | 58,752 | 9,504 | 50,272 | 793,424 | 822,018,208 | 49248 |
| | H=192 | 108,000 | 9,504 | 99,520 | 842,672 | 822,804,736 | 98496 |
| | H=336 | 181,872 | 9,504 | 173,392 | 916,544 | 823,984,528 | 172368 |
| | H=720 | 378,864 | 9,504 | 369,904 | 1,113,536 | 827,130,640 | 369360 |
| Relative Size (Avg) | | 1.0 | 0.039 | 0.953 | 5.040 | 4530.574 | 0.948 |

that both PSFormer (full) and SAMformer have parameter sizes that are close to RLinear, where RLinear's parameters mainly stem from the linear mapping between input and output. Notably, the parameter sizes of these three models are relatively unaffected by the number of input channels. In contrast, TSMixer and ModernTCN exhibit significantly larger parameter sizes, with the number of input channels playing a major role in the overall parameter burden. The relative size comparison shows that TSMixer and ModernTCN have several times, or even thousands of times, more parameters than PSFormer(full). Finally, the parameter size of PSFormer(Encoder) is much smaller, indicating that optimizing the linear mapping layer in the output could further reduce the overall parameter count.

## A.5   Running Time Comparison

The average running time per iteration (s/iter) of different models on the ETTh1 and Weather datasets with varying prediction horizons is shown in Table 10. PSformer demonstrates relatively stable running times across different horizons. For the ETTh1 dataset, the running time remains between 0.011 and 0.012 seconds, while for the Weather dataset, it varies slightly between 0.026 and 0.027 seconds. PSformer also shows comparatively efficient running times across the datasets, with performance that remains competitive even as the prediction horizon increases. This indicates that PSformer manages computational costs effectively, especially when compared to more complex models.

Table 10: Comparison of the running time (s/iter). We test the average running time per iteration of different models across the first five epochs on the ETTh1 and Weather datasets.

| Dataset | Horizon | PSformer | PatchTST | ModernTCN | TSMixer | RLinear | iTransformer |
|---------|---------|----------|----------|-----------|---------|---------|--------------|
| ETTh1 | H=96 | 0.012 | 0.049 | 0.241 | 0.013 | 0.032 | 0.020 |
| | H=192 | 0.011 | 0.049 | 0.244 | 0.016 | 0.033 | 0.023 |
| | H=336 | 0.012 | 0.054 | 0.245 | 0.015 | 0.034 | 0.022 |
| | H=720 | 0.012 | 0.063 | 0.279 | 0.019 | 0.037 | 0.025 |
| Weather | H=96 | 0.026 | 0.165 | 0.388 | 0.013 | 0.030 | 0.022 |
| | H=192 | 0.026 | 0.166 | 0.361 | 0.016 | 0.032 | 0.022 |
| | H=336 | 0.027 | 0.176 | 0.730 | 0.016 | 0.033 | 0.027 |
| | H=720 | 0.027 | 0.185 | 0.788 | 0.024 | 0.034 | 0.035 |

Table 11: Performance comparison with different positional encoding methods.

| Dataset | Horizon | Pos emb (time series) | Pos emb (segment) | No pos emb |
|---------|---------|-----------------------|-------------------|------------|
| **ETTh1** | 96 | 0.378 | 0.353 | 0.352 |
| | 192 | 0.412 | 0.388 | 0.385 |
| **ETTh2** | 96 | 0.298 | 0.276 | 0.272 |
| | 192 | 0.352 | 0.338 | 0.335 |
| **Exchange Rate** | 96 | 0.189 | 0.095 | 0.091 |
| | 192 | 0.314 | 0.201 | 0.197 |
| | 336 | 0.525 | 0.370 | 0.345 |
| | 720 | 1.574 | 1.041 | 1.036 |

## A.6 Discuss about Positional Encoding

**The Reasons of Eliminating Positional Encoding** On the one hand, the SAMformer does not use positional encoding because it applies attention to the channel dimension, where there is no strict sequential relationship between channels. Although SSA in PSformer is also a cross-channel structure, it involves local time series constructed by patches. We consider such local sequences as local representations (or tokens) rather than short sequences. The experimental results also demonstrate that this structure can similarly reduce the dependency on positional encoding while achieving good performance.

On the other hand, we conducted a set of comparative experiments using positional encoding to better illustrate its impact. Specifically, we tested the encoding performance under two different data transformation modes: pos emb (time series), where positional encoding is applied to the original time series before dimension transformation; and pos emb (segment), where positional encoding is applied to the transformed segments. The default case, No pos emb, refers to the absence of positional encoding.

To highlight the differences between seasonal vs. non-seasonal and stationary vs. non-stationary characteristics, we selected the ETTh1 and ETTh2 datasets (relatively seasonal and stable) as well as the Exchange dataset (relatively non-seasonal and non-stationary). The experimental results are shown in the Table 11.

The degraded performance of pos emb (time series) might be due to the incompatibility of the positional encoding with dimension transformation, as the original temporal order is lost in the segment dimension, making it unsuitable for dot-product attention calculations. On the other hand, pos emb (segment) shows smaller changes compared to the No pos emb case, but the performance still deteriorates slightly. This suggests that the significance of positional encoding in the context of multivariate time series forecasting might need to be re-evaluated, as there are fundamental differences between NLP and time series data when applying attention mechanisms.

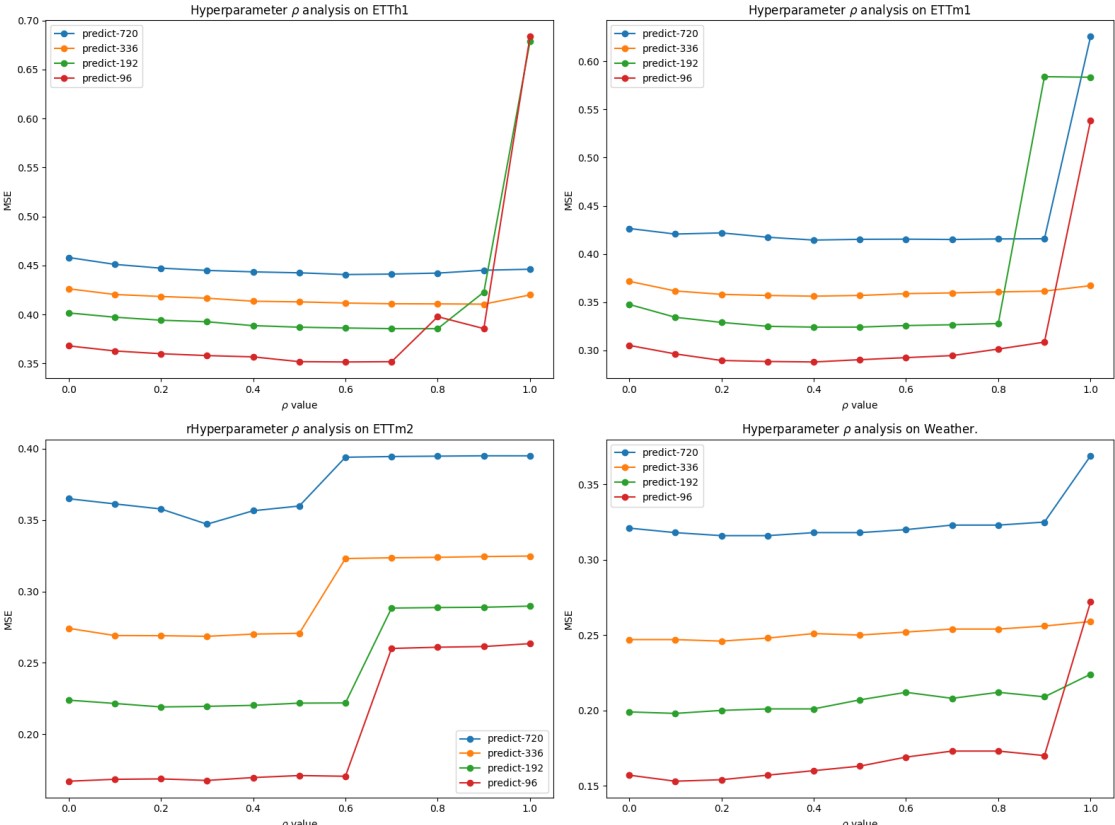

Figure 2: Ablation analysis on hyper-parameter $\rho$. When taking $\rho$ values from 0 to 1 in steps of 0.1, the prediction loss will slightly decrease first and then increase significantly if the $\rho$ exceeds a threshold, which means the selection of $\rho$ should be careful.

### A.7 Sharpness-Aware Minimization

**Optimization steps** SAM optimizer Foret et al. (2021) modifies the typical gradient descent update to seek a flatten optimum. Below is the mathematical formulation:

Let $\theta$ be the model parameters, $\mathcal{L}(\theta)$ be the loss function, and $\epsilon$ be a small perturbation applied to the parameters. The SAM optimization process can be described in two steps:

- Find the adversarial perturbation that maximizes the loss in a neighborhood of the current weights $\theta$:

$$\hat{\epsilon} = \arg \max_{\|\epsilon\| \leq \rho} \mathcal{L}(\theta + \epsilon)$$

where $\rho$ is a small constant that controls the size of the neighborhood.

- Update the parameters in the direction that minimizes the loss with respect to the perturbed parameters:

$$\theta \leftarrow \theta - \eta \nabla_\theta \mathcal{L}(\theta + \hat{\epsilon})$$

where $\eta$ is the learning rate.

As used in SAMformer, we also employ this optimization technique to train our models, which can generate promising results.

Table 12: Neighborhood size $\rho^*$ used by *PSformer*, *SAMformer* and *TSMixer* for SAM optimization to achieve their best performance on the benchmarks.

| H | Model | ETTh1 | ETTh2 | ETTm1 | ETTm2 | Electricity | Exchange | Traffic | Weather |
|---|---|---|---|---|---|---|---|---|---|
| 96 | *PSformer* | 0.6 | 0.1 | 0.4 | 0.0 | 0.0 | 0.2 | 0.1 | 0.1 |
| | *SAMformer* | 0.5 | 0.5 | 0.6 | 0.2 | 0.5 | 0.7 | 0.8 | 0.4 |
| | *TSMixer* | 1.0 | 0.9 | 1.0 | 1.0 | 1.0 | 0.9 | 0.0 | 0.5 |
| 192 | *PSformer* | 0.8 | 0.0 | 0.4 | 0.2 | 0.1 | 0.1 | 0.1 | 0.1 |
| | *SAMformer* | 0.6 | 0.8 | 0.9 | 0.9 | 0.6 | 0.8 | 0.1 | 0.4 |
| | *TSMixer* | 0.7 | 0.1 | 0.6 | 0.9 | 0.6 | 0.9 | 0.9 | 0.4 |
| 336 | *PSformer* | 0.9 | 0.6 | 0.4 | 0.3 | 0.1 | 0.2 | 0.2 | 0.2 |
| | *SAMformer* | 0.9 | 0.6 | 0.9 | 0.8 | 0.5 | 0.5 | 0.5 | 0.6 |
| | *TSMixer* | 0.7 | 0.7 | 0.9 | 1.0 | 0.4 | 1.0 | 0.6 | 0.6 |
| 720 | *PSformer* | 0.6 | 0.5 | 0.4 | 0.3 | 0.1 | 0.2 | 0.3 | 0.3 |
| | *SAMformer* | 0.9 | 0.8 | 0.9 | 0.9 | 1.0 | 0.9 | 0.7 | 0.5 |
| | *TSMixer* | 0.3 | 0.4 | 0.5 | 1.0 | 0.9 | 0.1 | 0.9 | 0.3 |

# B    More Results and Analysis

## B.1    Investigation of Hyper-parameter $\rho$

**The Effect of $\rho$.** Since we employed SAM to ensure training stability, we also conducted sensitivity tests on the hyper-parameter $\rho$ in SAM. We divided the range of $\rho$ from 0 to 1 into 10 equal parts and tested its effect on model prediction performance across the ETTh1, ETTm1, ETTm2, and Weather datasets. The results are shown in Figure 2. It can be observed that as the parameter $\rho$ gradually increases in SAM can improve the model's prediction performance to some extent. However, if $\rho$ is set too large, it may degrade the model's performance. When selecting $\rho$, it's important to consider the dataset's complexity and noise levels, as well as the model's architecture. For more complex datasets or larger models, a slightly larger $\rho$ can help smooth the loss landscape and improve generalization. Further, $\rho$ should also be balanced with the learning rate to avoid instability or performance degradation. As a comparison, we also report the $\rho^*$ used by *PSformer*, *SAMformer* and *TSMixer* in the Table 12.

## B.2    Full Results

Table 13 presents the detailed experimental results of different models and forecasting horizons, providing a comprehensive evaluation of their performance in long-term time series forecasting tasks. The performance is measured using Mean Squared Error (MSE) and Mean Absolute Error (MAE). In the table, red values represent the best performance in the respective task and metric, while blue-lined values indicate the second-best performance.

The last row of the table summarizes the number of first-place results for each model across all tasks. As can be seen, PSformer achieved the best MSE performance in 20 out of 32 prediction tasks, and ranked second in 8 tasks. In terms of MAE, PSformer achieved the best results in 22 tasks and came second in 5 tasks. This clearly demonstrates the superior performance of PSformer compared to other baseline models in long-term time series forecasting tasks.

The next best-performing model is ModernTCN, which achieved the best MSE results in 6 tasks and the best MAE results in 3 tasks. While other models such as SAMformer and PatchTST also showed competitive performance in some tasks, their overall results are not as strong as those of PSformer and ModernTCN. In summary, PSformer's strong performance across multiple benchmark tasks suggests its potential effectiveness in long-term forecasting. The statistical fluctuations across multiple random seeds are shown in Appendix B.2.

## B.3    Training Loss

Figure 3 illustrates the training and validation loss curves of the ETTh1 and ETTm1 datasets with prediction horizons $H = \{96, 192\}$. In this experiment, we set the number of epochs to 200 and disabled early stopping

Table 13: Full long-term forecasting results. We set the forecasting horizons $H \in \{96, 192, 336, 720\}$. A lower value indicates better performance. Avg means the average results from all prediction lengths. Red: the best, Blue lined: the second best.

| Models | | PSformer | | SAMformer | | TSMixer | | PatchTST | | MOMENT | | ModernTCN | | FEDformer | | GPT4TS | | Autoformer | | RLinear | | iTransformer | |
|---|---|---|---|---|---|---|---|---|---|---|---|---|---|---|---|---|---|---|---|---|---|---|---|
| Metric | | MSE | MAE | MSE | MAE | MSE | MAE | MSE | MAE | MSE | MAE | MSE | MAE | MSE | MAE | MSE | MAE | MSE | MAE | MSE | MAE | MSE | MAE |
| ETTh1 | 96 | 0.352 | 0.385 | 0.381 | 0.402 | 0.388 | 0.408 | 0.414 | 0.419 | 0.387 | 0.410 | 0.373 | 0.399 | 0.376 | 0.415 | 0.376 | 0.397 | 0.435 | 0.446 | 0.386 | 0.395 | 0.386 | 0.405 |
| | 192 | 0.385 | 0.406 | 0.409 | 0.418 | 0.421 | 0.426 | 0.460 | 0.445 | 0.410 | 0.426 | 0.407 | 0.419 | 0.423 | 0.446 | 0.416 | 0.418 | 0.456 | 0.457 | 0.437 | 0.424 | 0.441 | 0.436 |
| | 336 | 0.411 | 0.424 | 0.423 | 0.425 | 0.430 | 0.434 | 0.501 | 0.466 | 0.422 | 0.437 | 0.436 | 0.437 | 0.444 | 0.462 | 0.442 | 0.433 | 0.486 | 0.487 | 0.479 | 0.446 | 0.487 | 0.458 |
| | 720 | 0.440 | 0.456 | 0.427 | 0.449 | 0.440 | 0.459 | 0.500 | 0.488 | 0.454 | 0.472 | 0.467 | 0.474 | 0.469 | 0.492 | 0.477 | 0.515 | 0.515 | 0.517 | 0.481 | 0.470 | 0.503 | 0.491 |
| | Avg | 0.397 | 0.418 | 0.410 | 0.424 | 0.420 | 0.431 | 0.468 | 0.455 | 0.418 | 0.436 | 0.421 | 0.432 | 0.428 | 0.454 | 0.428 | 0.426 | 0.473 | 0.477 | 0.446 | 0.434 | 0.454 | 0.448 |
| ETTh2 | 96 | 0.272 | 0.337 | 0.295 | 0.358 | 0.305 | 0.367 | 0.302 | 0.348 | 0.288 | 0.345 | 0.271 | 0.339 | 0.332 | 0.374 | 0.285 | 0.342 | 0.332 | 0.368 | 0.288 | 0.338 | 0.297 | 0.349 |
| | 192 | 0.335 | 0.379 | 0.340 | 0.386 | 0.350 | 0.393 | 0.388 | 0.400 | 0.349 | 0.386 | 0.332 | 0.382 | 0.407 | 0.446 | 0.354 | 0.389 | 0.426 | 0.434 | 0.374 | 0.390 | 0.380 | 0.400 |
| | 336 | 0.356 | 0.411 | 0.350 | 0.395 | 0.360 | 0.404 | 0.426 | 0.433 | 0.369 | 0.408 | 0.365 | 0.411 | 0.400 | 0.447 | 0.373 | 0.407 | 0.477 | 0.479 | 0.415 | 0.426 | 0.428 | 0.432 |
| | 720 | 0.389 | 0.431 | 0.391 | 0.428 | 0.402 | 0.435 | 0.431 | 0.446 | 0.403 | 0.439 | 0.402 | 0.441 | 0.412 | 0.469 | 0.406 | 0.441 | 0.453 | 0.490 | 0.420 | 0.440 | 0.427 | 0.445 |
| | Avg | 0.338 | 0.390 | 0.344 | 0.391 | 0.354 | 0.400 | 0.387 | 0.407 | 0.352 | 0.394 | 0.343 | 0.393 | 0.388 | 0.434 | 0.355 | 0.395 | 0.422 | 0.443 | 0.374 | 0.398 | 0.383 | 0.407 |
| ETTm1 | 96 | 0.282 | 0.336 | 0.329 | 0.363 | 0.327 | 0.363 | 0.329 | 0.367 | 0.293 | 0.349 | 0.310 | 0.356 | 0.326 | 0.390 | 0.292 | 0.346 | 0.510 | 0.492 | 0.355 | 0.376 | 0.334 | 0.368 |
| | 192 | 0.321 | 0.360 | 0.353 | 0.378 | 0.356 | 0.381 | 0.367 | 0.385 | 0.326 | 0.368 | 0.340 | 0.373 | 0.365 | 0.415 | 0.332 | 0.372 | 0.514 | 0.495 | 0.391 | 0.392 | 0.377 | 0.391 |
| | 336 | 0.352 | 0.380 | 0.382 | 0.394 | 0.387 | 0.397 | 0.399 | 0.410 | 0.352 | 0.384 | 0.373 | 0.392 | 0.392 | 0.425 | 0.366 | 0.394 | 0.510 | 0.492 | 0.424 | 0.415 | 0.426 | 0.420 |
| | 720 | 0.413 | 0.412 | 0.429 | 0.418 | 0.441 | 0.425 | 0.454 | 0.439 | 0.405 | 0.416 | 0.420 | 0.418 | 0.446 | 0.458 | 0.417 | 0.421 | 0.527 | 0.492 | 0.487 | 0.450 | 0.491 | 0.459 |
| | Avg | 0.342 | 0.372 | 0.373 | 0.388 | 0.378 | 0.392 | 0.387 | 0.400 | 0.344 | 0.379 | 0.361 | 0.384 | 0.382 | 0.422 | 0.351 | 0.383 | 0.515 | 0.493 | 0.414 | 0.407 | 0.407 | 0.410 |
| ETTm2 | 96 | 0.167 | 0.258 | 0.181 | 0.274 | 0.190 | 0.284 | 0.175 | 0.259 | 0.170 | 0.260 | 0.168 | 0.261 | 0.180 | 0.271 | 0.173 | 0.262 | 0.205 | 0.293 | 0.182 | 0.265 | 0.180 | 0.264 |
| | 192 | 0.219 | 0.292 | 0.233 | 0.306 | 0.250 | 0.320 | 0.241 | 0.302 | 0.227 | 0.297 | 0.231 | 0.305 | 0.252 | 0.318 | 0.229 | 0.301 | 0.278 | 0.336 | 0.246 | 0.304 | 0.250 | 0.309 |
| | 336 | 0.269 | 0.325 | 0.285 | 0.338 | 0.301 | 0.350 | 0.305 | 0.343 | 0.275 | 0.328 | 0.272 | 0.328 | 0.324 | 0.364 | 0.286 | 0.341 | 0.343 | 0.379 | 0.307 | 0.342 | 0.311 | 0.348 |
| | 720 | 0.347 | 0.376 | 0.375 | 0.390 | 0.389 | 0.402 | 0.402 | 0.400 | 0.363 | 0.387 | 0.375 | 0.394 | 0.410 | 0.420 | 0.378 | 0.401 | 0.414 | 0.419 | 0.407 | 0.398 | 0.412 | 0.407 |
| | Avg | 0.251 | 0.313 | 0.269 | 0.327 | 0.283 | 0.339 | 0.281 | 0.326 | 0.259 | 0.318 | 0.262 | 0.322 | 0.292 | 0.343 | 0.267 | 0.326 | 0.310 | 0.357 | 0.286 | 0.327 | 0.288 | 0.332 |
| Weather | 96 | 0.149 | 0.200 | 0.197 | 0.249 | 0.189 | 0.242 | 0.177 | 0.218 | 0.154 | 0.209 | 0.154 | 0.209 | 0.238 | 0.314 | 0.162 | 0.212 | 0.249 | 0.329 | 0.192 | 0.232 | 0.174 | 0.214 |
| | 192 | 0.193 | 0.243 | 0.235 | 0.277 | 0.228 | 0.272 | 0.225 | 0.259 | 0.197 | 0.248 | 0.207 | 0.257 | 0.275 | 0.329 | 0.204 | 0.248 | 0.325 | 0.370 | 0.240 | 0.271 | 0.221 | 0.254 |
| | 336 | 0.245 | 0.282 | 0.276 | 0.304 | 0.271 | 0.299 | 0.278 | 0.297 | 0.246 | 0.285 | 0.248 | 0.289 | 0.339 | 0.377 | 0.254 | 0.286 | 0.351 | 0.391 | 0.292 | 0.307 | 0.278 | 0.296 |
| | 720 | 0.314 | 0.332 | 0.334 | 0.342 | 0.331 | 0.341 | 0.354 | 0.348 | 0.315 | 0.336 | 0.337 | 0.342 | 0.389 | 0.409 | 0.326 | 0.337 | 0.415 | 0.426 | 0.364 | 0.353 | 0.358 | 0.347 |
| | Avg | 0.225 | 0.264 | 0.261 | 0.293 | 0.255 | 0.289 | 0.259 | 0.281 | 0.228 | 0.270 | 0.237 | 0.274 | 0.310 | 0.357 | 0.237 | 0.271 | 0.335 | 0.379 | 0.272 | 0.291 | 0.258 | 0.278 |
| Electricity | 96 | 0.133 | 0.229 | 0.155 | 0.252 | 0.171 | 0.273 | 0.181 | 0.270 | 0.136 | 0.233 | 0.133 | 0.228 | 0.186 | 0.302 | 0.139 | 0.238 | 0.196 | 0.313 | 0.201 | 0.281 | 0.148 | 0.240 |
| | 192 | 0.149 | 0.242 | 0.168 | 0.263 | 0.191 | 0.292 | 0.188 | 0.274 | 0.152 | 0.247 | 0.147 | 0.241 | 0.197 | 0.311 | 0.153 | 0.251 | 0.211 | 0.324 | 0.201 | 0.283 | 0.162 | 0.253 |
| | 336 | 0.164 | 0.258 | 0.183 | 0.277 | 0.198 | 0.297 | 0.204 | 0.293 | 0.167 | 0.264 | 0.163 | 0.260 | 0.213 | 0.328 | 0.169 | 0.266 | 0.214 | 0.327 | 0.215 | 0.298 | 0.178 | 0.269 |
| | 720 | 0.203 | 0.291 | 0.219 | 0.306 | 0.230 | 0.321 | 0.246 | 0.324 | 0.205 | 0.295 | 0.194 | 0.289 | 0.233 | 0.344 | 0.206 | 0.297 | 0.236 | 0.342 | 0.257 | 0.331 | 0.225 | 0.317 |
| | Avg | 0.162 | 0.255 | 0.181 | 0.275 | 0.198 | 0.296 | 0.205 | 0.290 | 0.165 | 0.260 | 0.160 | 0.255 | 0.207 | 0.321 | 0.167 | 0.263 | 0.214 | 0.327 | 0.219 | 0.298 | 0.178 | 0.270 |
| Exchange | 96 | 0.081 | 0.197 | 0.161 | 0.306 | 0.233 | 0.363 | 0.088 | 0.205 | 0.098 | 0.224 | 0.207 | 0.342 | 0.139 | 0.276 | 0.091 | 0.212 | 0.197 | 0.323 | 0.093 | 0.217 | 0.086 | 0.206 |
| | 192 | 0.179 | 0.299 | 0.246 | 0.371 | 0.342 | 0.437 | 0.176 | 0.299 | 0.201 | 0.323 | 0.337 | 0.437 | 0.256 | 0.369 | 0.183 | 0.304 | 0.300 | 0.369 | 0.184 | 0.307 | 0.177 | 0.299 |
| | 336 | 0.328 | 0.412 | 0.368 | 0.453 | 0.474 | 0.515 | 0.301 | 0.397 | 0.387 | 0.454 | 0.520 | 0.553 | 0.426 | 0.464 | 0.328 | 0.417 | 0.509 | 0.524 | 0.351 | 0.432 | 0.331 | 0.417 |
| | 720 | 0.842 | 0.689 | 1.003 | 0.750 | 1.078 | 0.777 | 0.901 | 0.714 | 1.062 | 0.783 | 1.154 | 0.810 | 1.090 | 0.800 | 0.880 | 0.704 | 1.447 | 0.941 | 0.886 | 0.714 | 0.847 | 0.691 |
| | Avg | 0.358 | 0.399 | 0.445 | 0.470 | 0.532 | 0.523 | 0.367 | 0.404 | 0.437 | 0.446 | 0.555 | 0.536 | 0.478 | 0.477 | 0.371 | 0.409 | 0.613 | 0.539 | 0.378 | 0.417 | 0.360 | 0.403 |
| Traffic | 96 | 0.367 | 0.257 | 0.407 | 0.292 | 0.409 | 0.300 | 0.462 | 0.295 | 0.391 | 0.282 | 0.391 | 0.271 | 0.576 | 0.359 | 0.388 | 0.282 | 0.597 | 0.371 | 0.649 | 0.389 | 0.395 | 0.268 |
| | 192 | 0.390 | 0.272 | 0.415 | 0.294 | 0.433 | 0.317 | 0.466 | 0.296 | 0.404 | 0.287 | 0.403 | 0.275 | 0.610 | 0.380 | 0.407 | 0.290 | 0.607 | 0.382 | 0.601 | 0.366 | 0.417 | 0.276 |
| | 336 | 0.404 | 0.274 | 0.421 | 0.292 | 0.424 | 0.299 | 0.482 | 0.304 | 0.414 | 0.292 | 0.410 | 0.280 | 0.608 | 0.375 | 0.412 | 0.294 | 0.623 | 0.387 | 0.609 | 0.369 | 0.433 | 0.283 |
| | 720 | 0.439 | 0.294 | 0.456 | 0.311 | 0.488 | 0.344 | 0.514 | 0.322 | 0.450 | 0.310 | 0.451 | 0.305 | 0.621 | 0.375 | 0.450 | 0.312 | 0.639 | 0.395 | 0.647 | 0.387 | 0.467 | 0.302 |
| | Avg | 0.400 | 0.274 | 0.425 | 0.297 | 0.439 | 0.315 | 0.481 | 0.304 | 0.415 | 0.293 | 0.414 | 0.283 | 0.604 | 0.372 | 0.414 | 0.295 | 0.617 | 0.384 | 0.626 | 0.378 | 0.428 | 0.282 |
| $1^{st}$ Count | | 20 | 22 | 2 | 3 | 0 | 0 | 2 | 3 | 2 | 0 | 6 | 3 | 0 | 0 | 0 | 0 | 0 | 0 | 0 | 0 | 2 | 2 |

Table 14: Statistical significance tests. Each task with average mean and standard deviation executed under 5 runs with different random seeds.

| Metric | Length | ETTh1 | ETTh2 | ETTm1 | ETTm2 | Weather | Exchange |
|--------|--------|-------|-------|-------|-------|---------|----------|
| **MSE** | 96 | 0.352±0.001 | 0.272±0.002 | 0.283±0.001 | 0.169±0.001 | 0.150±0.001 | 0.081±0.000 |
| | 192 | 0.386±0.001 | 0.338±0.003 | 0.322±0.001 | 0.219±0.001 | 0.195±0.001 | 0.177±0.004 |
| | 336 | 0.412±0.003 | 0.356±0.001 | 0.353±0.002 | 0.269±0.001 | 0.243±0.001 | 0.321±0.004 |
| | 720 | 0.443±0.002 | 0.390±0.001 | 0.414±0.001 | 0.352±0.004 | 0.314±0.001 | 0.840±0.007 |
| | Average | 0.398±0.002 | 0.339±0.002 | 0.343±0.001 | 0.252±0.002 | 0.225±0.001 | 0.355±0.004 |
| **MAE** | 96 | 0.385±0.001 | 0.336±0.001 | 0.342±0.001 | 0.259±0.001 | 0.201±0.001 | 0.197±0.000 |
| | 192 | 0.406±0.001 | 0.380±0.002 | 0.362±0.001 | 0.292±0.001 | 0.244±0.001 | 0.299±0.003 |
| | 336 | 0.424±0.004 | 0.409±0.003 | 0.380±0.001 | 0.325±0.001 | 0.280±0.001 | 0.408±0.003 |
| | 720 | 0.459±0.001 | 0.432±0.001 | 0.412±0.001 | 0.377±0.002 | 0.332±0.001 | 0.687±0.002 |
| | Average | 0.419±0.002 | 0.389±0.002 | 0.374±0.001 | 0.313±0.001 | 0.264±0.001 | 0.398±0.002 |

to observe the complete training process. As shown in the plots, despite the model reaching convergence early in the training (as indicated by the flattening of the training loss curve), the validation loss remains consistently low throughout the training duration. This indicates the model's stability and its ability to generalize well to unseen data over extended epochs.

### B.4 Visualization

To better understand our method, we present the forecast curves and the corresponding attention maps for several selected samples. The attention maps visualize the attention weights in different stages of the model, providing insight into how the model processes the input data at each stage.

For the ETTh1 in Figure 4, the input sequence length is 512, and we display the last 100 time steps of the input. The prediction length is set to 96. We select five samples from the ETTh1 dataset, and for each sample, we visualize the attention maps for stage 1 and stage 2 of the SSA. From the attention maps, it is evident that there are significant variations in the attention distributions across different samples. Additionally, the attention maps from stage 1 and stage 2 also show noticeable differences, despite both stages sharing the same PS block parameters. This indicates that while the two-stage share parameters, they are able to handle and process the information differently, capturing different aspects of the input data at each stage of the model.

For the Weather in Figure 5, the input length is also 512, and the last 100 time steps are displayed, while the prediction length is 192. Since the model for this dataset employs a three-layer Encoder structure, we display the attention maps for both stages across each layer. Specifically, the notation"1-2" represents the attention map for layer 1, stage 2, and similarly for the other layers. The first two rows of attention maps correspond to the attention distributions from the three Encoder layers. Following that, the prediction curves for nine selected variates are plotted, providing a detailed view of the model's forecast performance across different variates.

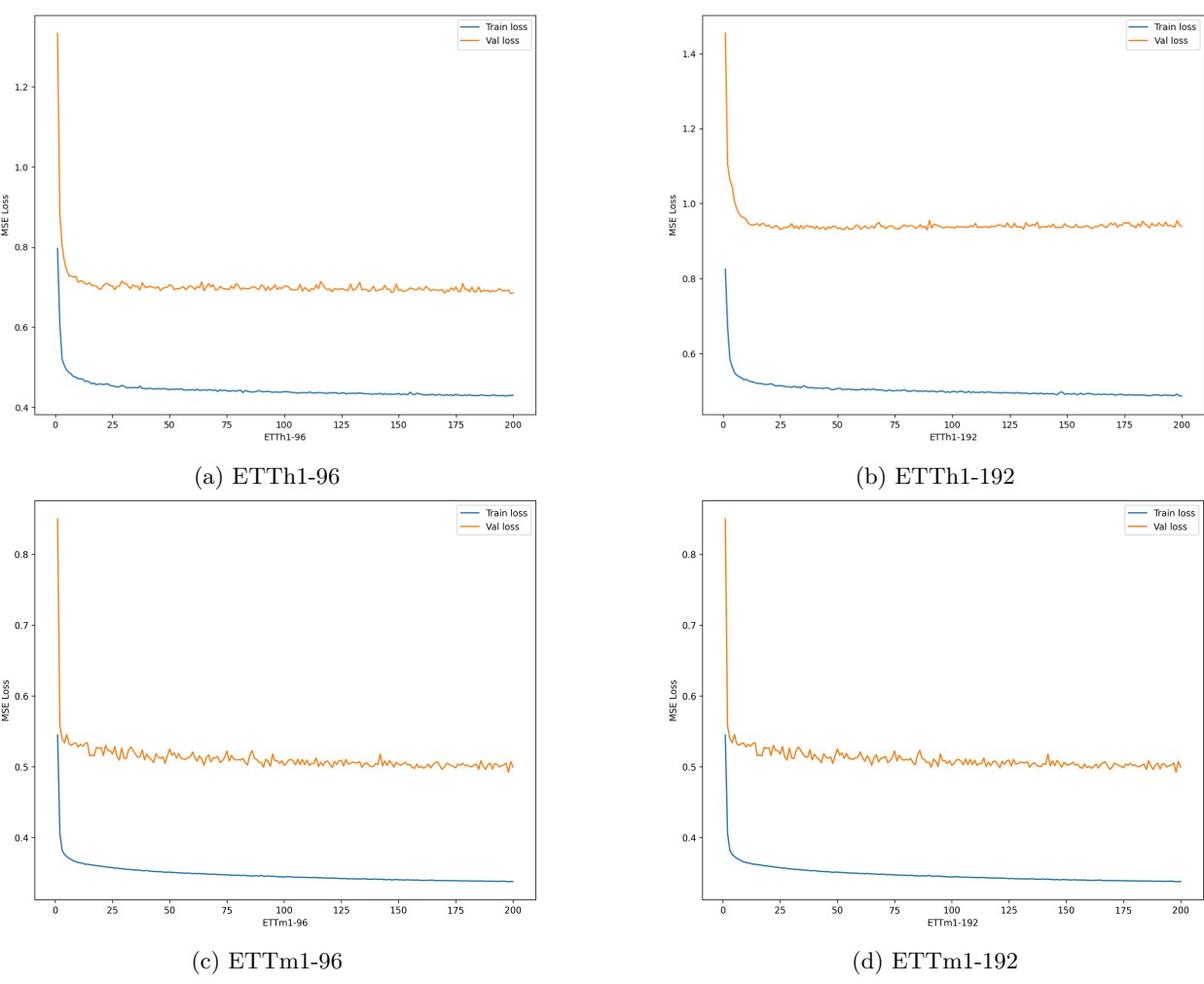

(a) ETTh1-96

(b) ETTh1-192

(c) ETTm1-96

(d) ETTm1-192

Figure 3: Training and validation loss curves of the ETTh1 and ETTm1 datasets.

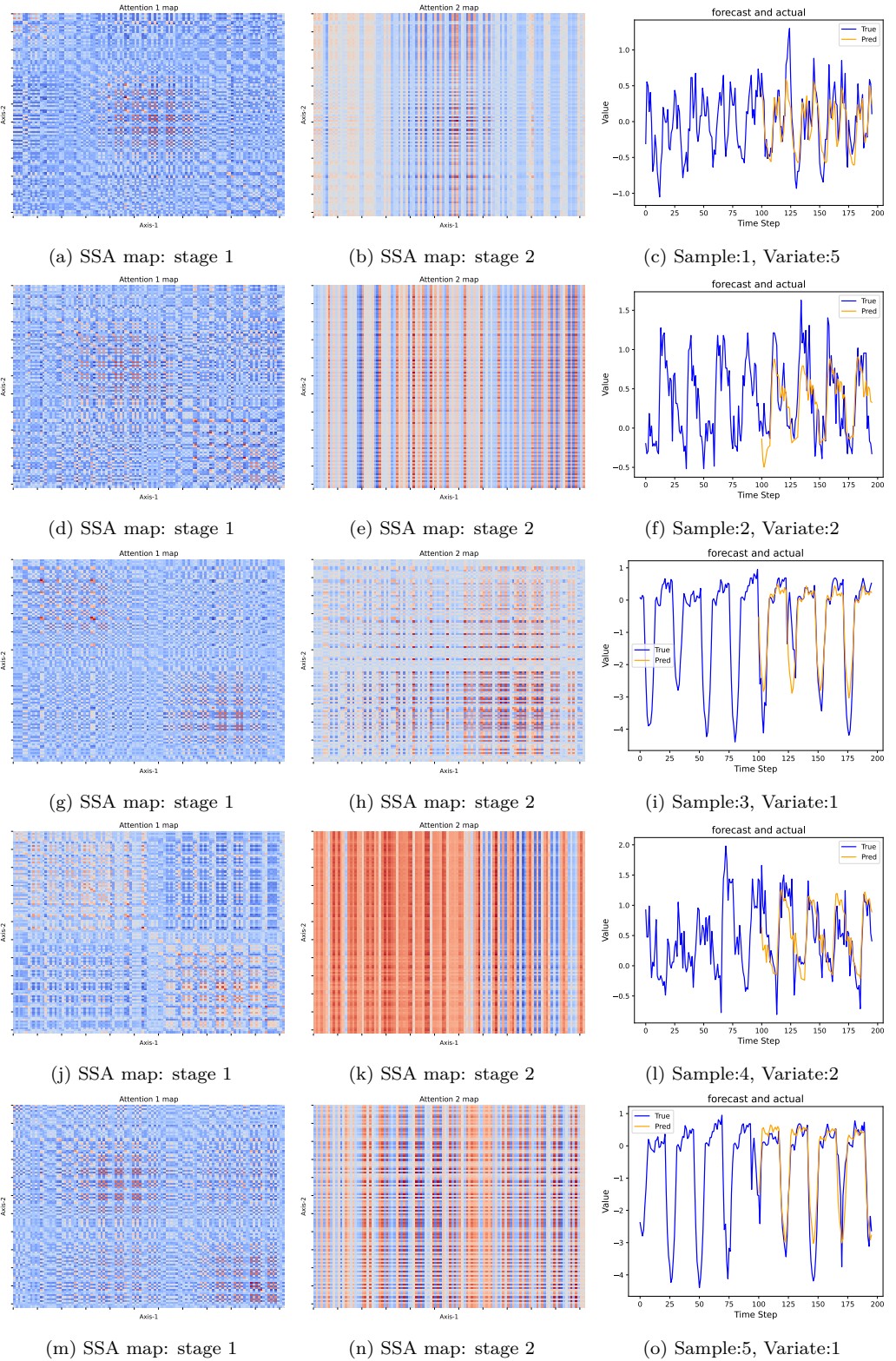

(a) SSA map: stage 1    (b) SSA map: stage 2    (c) Sample:1, Variate:5

(d) SSA map: stage 1    (e) SSA map: stage 2    (f) Sample:2, Variate:2

(g) SSA map: stage 1    (h) SSA map: stage 2    (i) Sample:3, Variate:1

(j) SSA map: stage 1    (k) SSA map: stage 2    (l) Sample:4, Variate:2

(m) SSA map: stage 1    (n) SSA map: stage 2    (o) Sample:5, Variate:1

Figure 4: SSA map and forecast samples for ETTh1-96

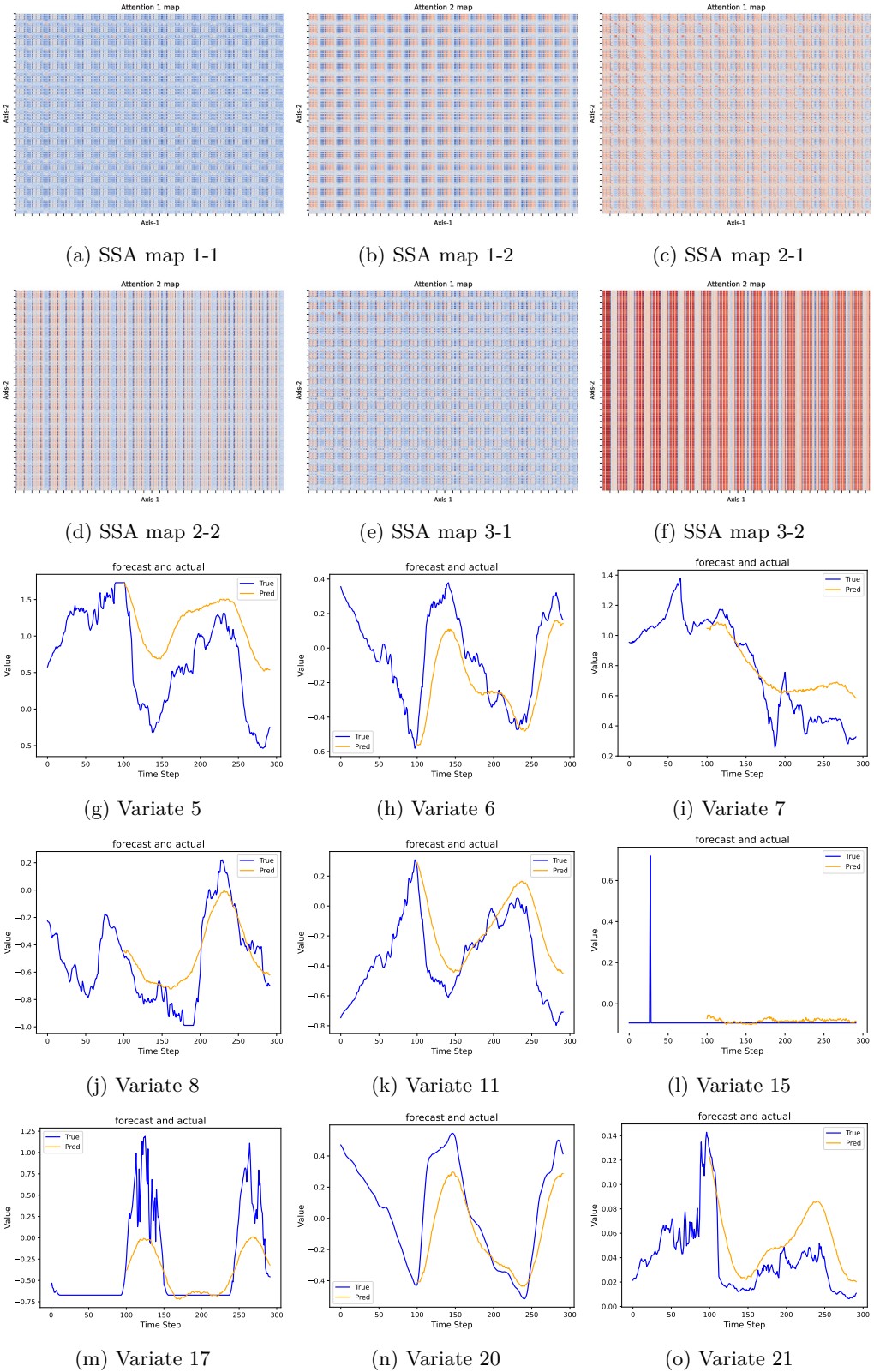

Figure 5: SSA matrices and forecast for Weather-192

## B.5   Additional Visualization of Attention Map

We further analyze the SSA attention matrix of PSformer in both the temporal and spatial dimensions. Using samples from ETTh1, the key parameters include: input length $L = 512$, segment number $N = 32$, channel number $M = 7$, and patch length $P = 16$. Therefore, $QK^T \in R^{C \times C}$, where $C = M \times P$, as described in the Spatial-Temporal Segment Attention part.

### B.5.1   SSA Attention Map

Since stage one of SSA operates on the input data x, we print out the attention matrix to observe how SSA captures local spatiotemporal information. The attention matrix is shown in the Figure 8 below. To better display the colors, we choose the plasma color style and apply a clipping operation to the top 1.

Since both the x-axis and y-axis correspond to a specific point within a segment of length C, the brighter yellow points in this attention matrix indicate higher attention weights between the corresponding x-axis and y-axis positions. Taking this attention map as an example, we can observe distinct high-attention regions (e.g., the top-left corner) and high-attention areas between different time steps and variables (e.g., the non-diagonal symmetric grid section in the top-left corner). From this, we can observe that the coordinate positions within the spatial attention sub-matrix (inside the grids) exhibit the same or gradually changing attention weights across different time steps (between the grids). This suggests that the model may capture temporal consistency between variables, local stationary features, and long-term dependencies across time steps.

### B.5.2   Single-channel SSA Attention Map

For better visualization of SSA's capture of the local temporal dimension, we extract the single-channel attention sub-matrix from the attention matrix, as shown in the upper corner of the Figure 7, and visualize the corresponding single-channel local time sequence in the lower corner of the Figure 7. This allows us to observe the high-attention weights at specific temporal local positions. We present three samples of attention along the temporal dimension for a single sequence.

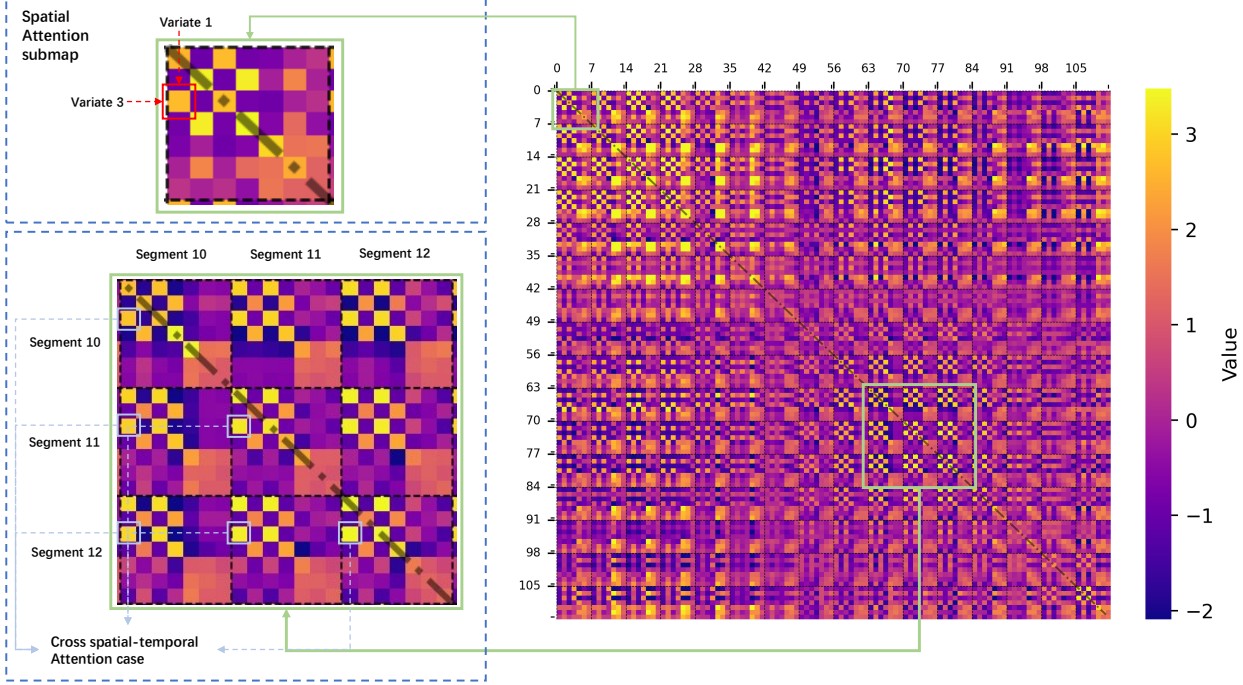

Figure 6: SSA attention map.

From the upper-left corner of Figure 7, the high attention weight at coordinates (x=7, y=14) corresponds to time steps 7 and 14 in the lower-left corner of Figure 7. These two time points represent a pattern of significant change in the sequence, which the model identifies as a high attention weight.

Using the same analysis method, we can observe in the middle column of the Figure 7 that high attention occurs between time steps 2 and 3, as well as between time steps 9 and 10. This includes both consecutive time points with related attention (e.g., 2 and 3) and attention with intervals (e.g., 2-3 and 9-10), which correspond to the upward phase and the stable phase of the time series.

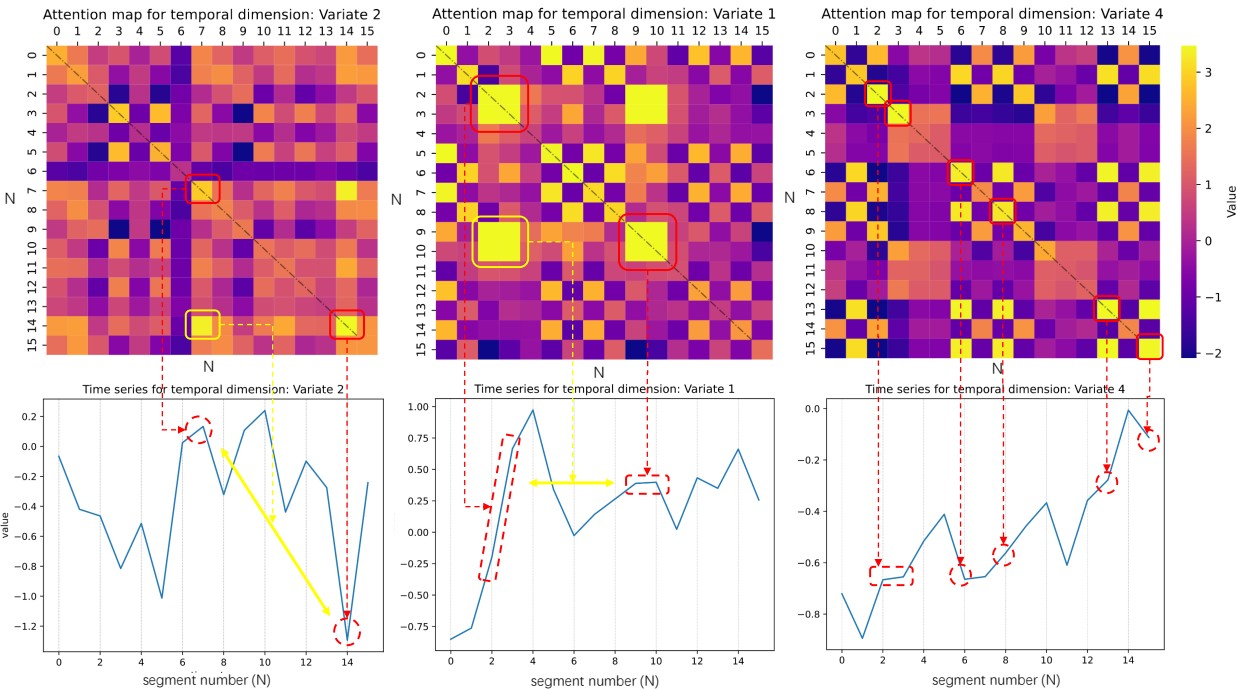

Figure 7: SSA intra-channel attention submap.

### B.5.3 Cross-channels SSA Attention Map

To visualize the cross-channel attention relationships in SSA, we extract the attention weights between two channels, forming the corresponding cross-channel attention sub-matrix map and the time series plots for the two variables, as shown in Figure 8.

From upper-left of the Figure 8, we can see that high attention is mainly concentrated at coordinates (3, 14) and (5, 14), which correspond to time step 14 of variate 2 and time steps 3 and 5 of variate 0. These three positions may reflect local minima for the two time series. In the middle column, the high attention weights correspond to large changes in opposite directions between variate 0 and variate 6. In the right column of Figure 8, we can see that the segments corresponding to segment numbers 1 and 2 of variate 5 exhibit higher attention values compared to the overall variate 0. Additionally, the attention on time steps 4 and 5, as well as 11 and 12, of variate 0 is higher.

### B.5.4 Additional Visualization of the SSA Attention Map without Parameter Sharing

We further analyzed how parameter sharing affects the model's ability to capture temporal patterns by comparing attention maps with and without parameter sharing. To achieve this, we also visualized the attention heatmaps without parameter sharing. To reflect cross-layer variations, we provided attention maps from both the first and second layers for comparison, as in Figure 9a and Figure 9b. Several differences were observed when comparing these to the attention maps with parameter sharing.

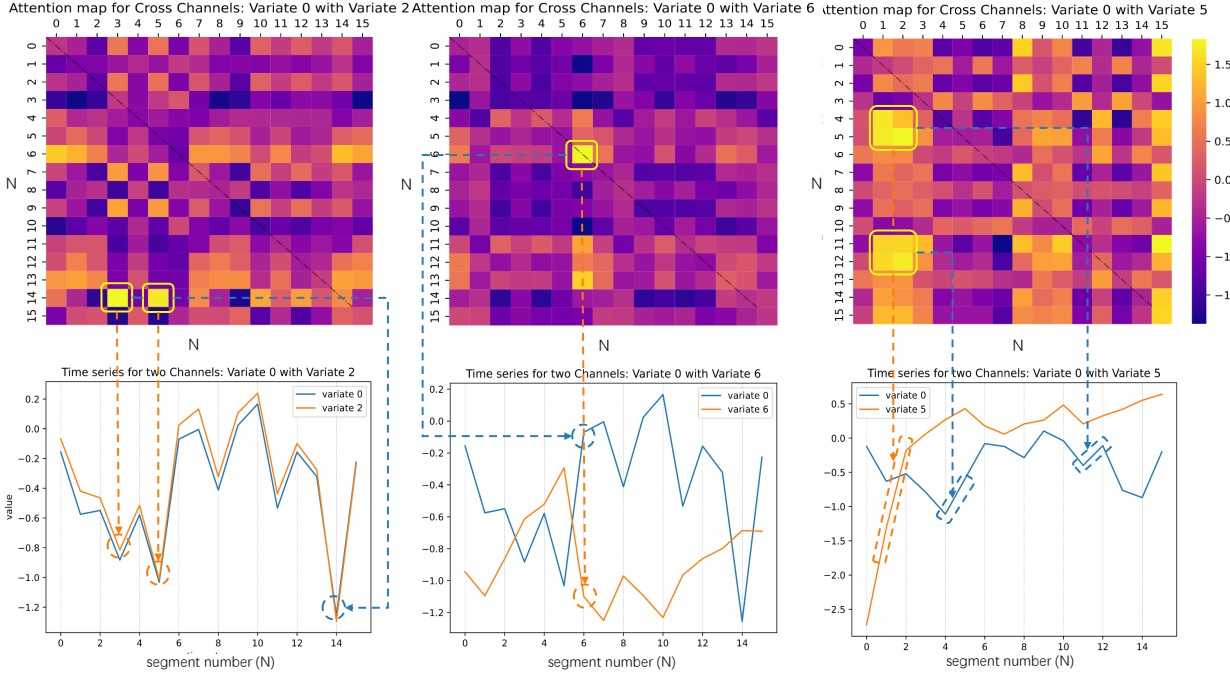

Figure 8: SSA cross-channel attention sub-map.

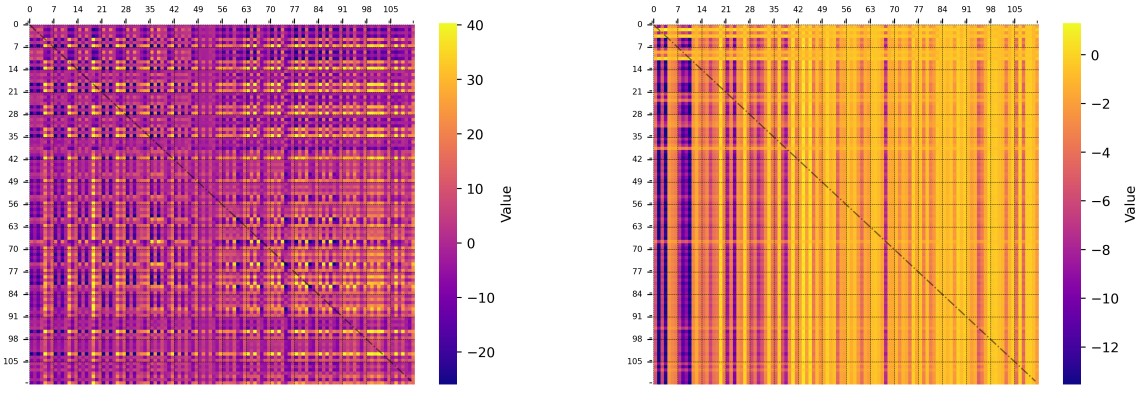

(a) Attention1 map without parameter sharing.  (b) Attention2 map without parameter sharing.

Figure 9: Comparison of Variate 2 Attention Maps

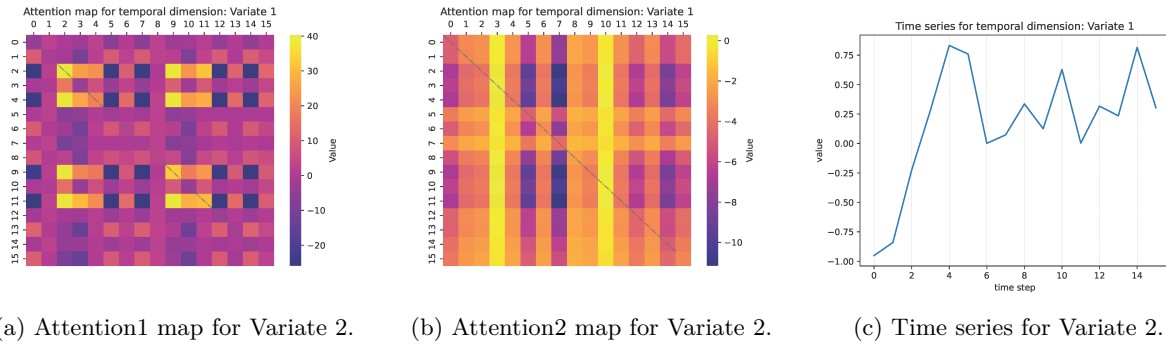

(a) Attention1 map for Variate 2.  (b) Attention2 map for Variate 2.  (c) Time series for Variate 2.

Figure 10: Single-channel Attention Map without Parameter Sharing.

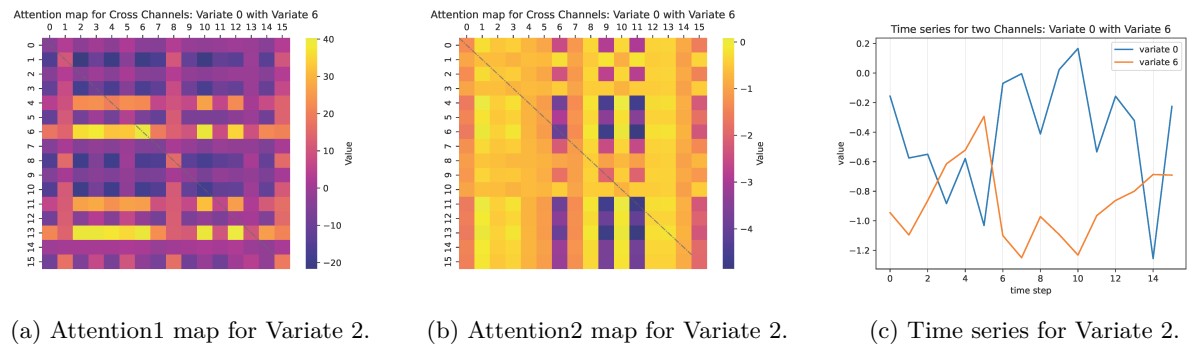

(a) Attention1 map for Variate 2.  (b) Attention2 map for Variate 2.  (c) Time series for Variate 2.

Figure 11: Comparison of Cross-channels Attention Maps without Parameter Sharing.

Range of attention weights: For attention maps with parameter sharing, the values generally range between [-3, 3], while for those without parameter sharing, the range is much broader, approximately [-30, 40]. The larger variations in attention weights without parameter sharing might contribute to faster model convergence during training.

Inter-channel relationships: With parameter sharing, the inter-channel relationships represented in Figure 10 are simpler and more distinct. Gradual transitions in attention weights between grid cells are clearly visible. In contrast, without parameter sharing, although progressive changes are still observed, the temporal relationships become more complex and harder to interpret (as the attention map in the first layer cannot be directly aligned with the corresponding temporal positions due to the lack of parameter sharing).

Cross-layer variations: Observing the differences between the two layers in the non-parameter-shared attention maps in Figure 11, I tend to suggest that the first layer's attention focuses on capturing basic temporal patterns, which are then refined and processed in the second layer. However, the interpretability in temporal models is challenging and remains an open research question worthy of further exploration.

### B.6 SSA for Univariate Time Series and Low Dependency Series

We tested the performance of PSformer in single-sequence forecasting. Specifically, we saved the 8 variables from the Exchange dataset into 8 separate single-sequence files, each supplemented with an additional column filled with zeros to form single sequences, along with an unrelated variable. We compared PSformer with the baseline model PatchTST (which is channel-independent and performs well on the Exchange dataset).

The experimental results in the Table 15. As can be observed, PSformer consistently outperforms across all uni-variate time series, demonstrating that the PSformer architecture is not only effective in capturing cross-channel information but also performs well in univariate time series or with little dependency between variables.

Table 15: Performance comparison between PSformer and PatchTST across various variates.

| Variate | Model | 96 | 192 | 336 | 720 |
|---|---|---|---|---|---|
| 1 | PSformer | 0.057 | 0.147 | 0.360 | 0.803 |
| | PatchTST | 0.063 | 0.152 | 0.461 | - |
| 2 | PSformer | 0.042 | 0.094 | 0.152 | 0.216 |
| | PatchTST | 0.052 | 0.140 | 0.181 | - |
| 3 | PSformer | 0.034 | 0.093 | 0.169 | 0.476 |
| | PatchTST | 0.055 | 0.116 | 0.176 | - |
| 4 | PSformer | 0.041 | 0.066 | 0.090 | 0.146 |
| | PatchTST | 0.058 | 0.085 | 0.111 | - |
| 5 | PSformer | 0.007 | 0.009 | 0.012 | 0.073 |
| | PatchTST | 0.016 | 0.025 | 0.022 | - |
| 6 | PSformer | 0.077 | 0.169 | 0.532 | 1.125 |
| | PatchTST | 0.099 | 0.215 | 0.505 | - |
| 7 | PSformer | 0.033 | 0.069 | 0.120 | 0.342 |
| | PatchTST | 0.039 | 0.079 | 0.140 | - |
| OT | PSformer | 0.047 | 0.101 | 0.190 | 0.510 |
| | PatchTST | 0.097 | 0.154 | 0.230 | - |

Table 16: Performance comparison with different norm window sizes on the Exchange Rate dataset.

| Dataset | Horizon | 16 | 64 | 128 | 256 | 512 |
|---|---|---|---|---|---|---|
| | 96 | 0.081 | 0.085 | 0.090 | 0.092 | 0.091 |
| | 192 | 0.179 | 0.187 | 0.189 | 0.191 | 0.197 |
| Exchange Rate | 336 | 0.328 | 0.338 | 0.356 | 0.362 | 0.345 |
| | 720 | 0.842 | 0.900 | 0.976 | 1.003 | 1.036 |
| | Avg | 0.358 | 0.378 | 0.403 | 0.412 | 0.417 |

During the experiments, PatchTST encountered a NaN loss on the validation set for a prediction length of 720, so the corresponding loss value was not recorded.

## B.7  Exchange Dateset Performance with Different RevIN Look-back Windows

The Exchange dataset is non-stationary in nature and has random walk characteristics, which prevent RevIN from obtaining stable mean and variance statistics. These statistics are sensitive to the choice of RevIN's look-back window. Further testing in the Table 16 revealed that the model performs best when the look-back window for calculating RevIN's statistics is very small (length 16), achieving results superior to all selected baseline models. We believe that for non-stationary data, RevIN's normalization should be used with caution. Adjusting the look-back window length can help identify more stable statistical means and variances, thereby facilitating model training.

## B.8  Comparison with Extended Datasets and Baselines

### B.8.1 Comparison with Additional Datasets

We also evaluated the performance of PSformer on the ILI and FRED-MD datasets and compared it with models such as TimeCMA Liu et al. (2025) and UniTime Liu et al. (2024a). The experimental results are presented in Table 17, where PSformer achieved the best performance on the ILI dataset, while TimeCMA performed better on the FRED-MD dataset. This may indicate that for different time series forecasting datasets, corresponding time series models should be adopted to achieve better performance.

Table 17: Comparison with additional datasets. We evaluate the model's performance on the ILI and FRED-MD datasets with forecasting horizons $H \in \{24, 36, 42, 60\}$, and the MSE and average loss are reported.

| | | PSformer | TimeCMA | Time-LLM | UniTime | GPT4TS | iTransformer | PatchTST | TimesNet | Dlinear | FEDformer |
|---|---|---|---|---|---|---|---|---|---|---|---|
| ILI | 24 | **1.893** | 1.996 | 2.383 | 2.346 | 2.732 | 2.347 | 2.335 | 2.317 | 2.398 | 3.228 |
| | 36 | **1.778** | 1.946 | 2.390 | 1.998 | 2.664 | 2.468 | 2.561 | 1.972 | 2.646 | 2.679 |
| | 48 | **1.851** | 1.940 | 2.394 | 1.979 | 2.617 | 2.489 | 2.465 | 2.238 | 2.614 | 2.622 |
| | 60 | **2.071** | 2.114 | 2.562 | 2.109 | 2.478 | 2.471 | 2.189 | 2.027 | 2.804 | 2.857 |
| | Avg | **1.898** | 1.999 | 2.432 | 2.108 | 2.623 | 2.444 | 2.388 | 2.139 | 2.616 | 2.847 |
| FRED-MD | 24 | 22.777 | **22.702** | 27.285 | 31.178 | 28.317 | 28.017 | 35.777 | 43.268 | 37.898 | 66.09 |
| | 36 | 48.679 | **41.792** | 48.730 | 54.172 | 59.520 | 50.837 | 61.034 | 69.514 | 71.047 | 94.359 |
| | 48 | 74.420 | **64.364** | 73.494 | 83.836 | 74.808 | 78.018 | 93.482 | 89.913 | 118.579 | 129.798 |
| | 60 | 108.256 | **77.792** | 108.221 | 118.429 | 83.613 | 90.212 | 133.444 | 116.187 | 156.844 | 173.616 |
| | Avg | 63.533 | **51.662** | 64.433 | 75.771 | 61.565 | 61.771 | 80.934 | 79.721 | 96.092 | 115.966 |

### B.8.2 Comparasion with Additional Baselines (Part-I)

We have collected the extended experimental MSE loss results of the relevant models in Table 18. Although there are differences in experimental setups across each work, which may affect the results and prevent a completely fair comparison, we have provided some key settings to help better understand the model performance.

From the results, compared to models with fixed windows, PSformer performed best on 7/8 of the prediction tasks. This further highlights PSformer's competitive performance in forecasting. Even when compared to non-fixed window models like FITS, PSformer performed best on 4/7 of the prediction tasks.

### B.8.3 Comparison with Additional Baselines (Part-II)

We have collected the extended experimental results of the relevant models in Table 19. Comparison with PDF and PatchTST. We set PDF and PatchTST with the same setting (input length to 512, $drop\_last = False$). The results provide further insights into PSformer performance.

We conducted tests on the following five datasets. From the experimental results, both PSformer and PDF significantly outperform PatchTST in predictive performance, with PSformer achieving better predictions than PDF. However, the average prediction loss reduction relative to PDF is not substantial. Therefore, we consider PSformer and PDF to exhibit equally excellent predictive performance under the same settings.

Comparison with Time-LLM. For Time-LLM, we listed the predictive performance from the original paper of the model in the table below. However, we checked its official repository and found Time-LLM also faces a DL issue in the test set dataloader, which may affect its reported results. Besides, due to the computational demands of large-scale models, we decided not to execute its code directly in our experiments. When selecting the baseline large models, we considered both MOMENT and TimeLLM. We ultimately chose MOMENT for two main reasons: the MOMENT paper includes a direct comparison with TimeLLM, and it is relatively lightweight (428M parameters). In summary, despite the significant difference in parameter scale and the DL issue present in TimeLLM, PSformer still achieves equal or better average predictive performance than TimeLLM on 3 out of 5 datasets.

Table 18: Comparasion with additional baselines (Part-I).

| | Models | PSformer | TimeMixer | CrossGNN | MICN | TimesNet | FITS |
|---|---|---|---|---|---|---|---|
| ETTh1 | **96** | 0.352 | 0.375 | 0.382 | \ | 0.384 | 0.372 |
| | **192** | 0.385 | 0.429 | 0.427 | \ | 0.436 | 0.404 |
| | **336** | 0.411 | 0.484 | 0.465 | \ | 0.491 | 0.427 |
| | **720** | 0.44 | 0.498 | 0.472 | \ | 0.521 | 0.424 |
| | **Avg** | 0.397 | 0.447 | 0.437 | \ | 0.458 | 0.407 |
| ETTh2 | **96** | 0.272 | 0.289 | 0.309 | \ | 0.34 | 0.271 |
| | **192** | 0.335 | 0.372 | 0.39 | \ | 0.402 | 0.331 |
| | **336** | 0.356 | 0.386 | 0.426 | \ | 0.452 | 0.354 |
| | **720** | 0.389 | 0.412 | 0.445 | \ | 0.462 | 0.377 |
| | **Avg** | 0.338 | 0.365 | 0.393 | \ | 0.414 | 0.333 |
| ETTm1 | **96** | 0.282 | 0.32 | 0.335 | \ | 0.338 | 0.303 |
| | **192** | 0.321 | 0.361 | 0.372 | \ | 0.372 | 0.337 |
| | **336** | 0.352 | 0.39 | 0.403 | \ | 0.41 | 0.366 |
| | **720** | 0.413 | 0.454 | 0.461 | \ | 0.478 | 0.415 |
| | **Avg** | 0.342 | 0.381 | 0.393 | \ | 0.4 | 0.355 |
| ETTm2 | **96** | 0.167 | 0.175 | 0.176 | 0.179 | 0.187 | 0.162 |
| | **192** | 0.219 | 0.237 | 0.24 | 0.307 | 0.249 | 0.216 |
| | **336** | 0.269 | 0.298 | 0.304 | 0.325 | 0.321 | 0.268 |
| | **720** | 0.347 | 0.391 | 0.406 | 0.502 | 0.408 | 0.348 |
| | **Avg** | 0.251 | 0.275 | 0.282 | 0.328 | 0.291 | 0.249 |
| Weather | **96** | 0.149 | 0.163 | 0.159 | \ | 0.172 | 0.143 |
| | **192** | 0.193 | 0.208 | 0.211 | \ | 0.219 | 0.186 |
| | **336** | 0.245 | 0.251 | 0.267 | \ | 0.28 | 0.236 |
| | **720** | 0.314 | 0.339 | 0.352 | \ | 0.365 | 0.307 |
| | **Avg** | 0.225 | 0.24 | 0.247 | \ | 0.259 | 0.218 |
| Electricity | **96** | 0.133 | 0.153 | 0.173 | 0.164 | 0.168 | 0.134 |
| | **192** | 0.149 | 0.166 | 0.195 | 0.177 | 0.184 | 0.149 |
| | **336** | 0.164 | 0.185 | 0.206 | 0.193 | 0.198 | 0.165 |
| | **720** | 0.203 | 0.225 | 0.231 | 0.212 | 0.22 | 0.203 |
| | **Avg** | 0.162 | 0.182 | 0.201 | 0.187 | 0.192 | 0.163 |
| Exchange rate | **96** | 0.091 | 0.09 | 0.084 | 0.102 | 0.107 | \ |
| | **192** | 0.197 | 0.187 | 0.171 | 0.172 | 0.226 | \ |
| | **336** | 0.345 | 0.353 | 0.319 | 0.272 | 0.367 | \ |
| | **720** | 1.036 | 0.934 | 0.805 | 0.714 | 0.964 | \ |
| | **Avg** | 0.417 | 0.391 | 0.345 | 0.315 | 0.416 | \ |
| Traffic | **96** | 0.367 | 0.462 | 0.57 | 0.519 | 0.593 | 0.385 |
| | **192** | 0.39 | 0.473 | 0.577 | 0.537 | 0.617 | 0.397 |
| | **336** | 0.404 | 0.498 | 0.588 | 0.534 | 0.629 | 0.41 |
| | **720** | 0.439 | 0.506 | 0.597 | 0.577 | 0.64 | 0.448 |
| | **Avg** | 0.4 | 0.485 | 0.583 | 0.542 | 0.62 | 0.41 |

Table 19: Comparison with additional baselines (Part-II).

|  | Models | PSformer | Crossformer | PDF | PatchTST | TimeLLM |
|---|---|---|---|---|---|---|
| ETTh1 | **96** | 0.352 | 0.384 | 0.361 | 0.374 | 0.362 |
|  | **192** | 0.385 | 0.438 | 0.391 | 0.413 | 0.398 |
|  | **336** | 0.411 | 0.495 | 0.415 | 0.434 | 0.43 |
|  | **720** | 0.44 | 0.522 | 0.468 | 0.455 | 0.442 |
|  | **Avg** | 0.397 | 0.46 | 0.409 | 0.419 | 0.408 |
| ETTh2 | **96** | 0.272 | 0.347 | 0.272 | 0.274 | 0.268 |
|  | **192** | 0.335 | 0.419 | 0.334 | 0.341 | 0.329 |
|  | **336** | 0.356 | 0.449 | 0.357 | 0.364 | 0.368 |
|  | **720** | 0.389 | 0.479 | 0.397 | 0.39 | 0.372 |
|  | **Avg** | 0.338 | 0.424 | 0.34 | 0.342 | 0.334 |
| ETTm1 | **96** | 0.282 | 0.349 | 0.284 | 0.29 | 0.272 |
|  | **192** | 0.321 | 0.405 | 0.327 | 0.333 | 0.31 |
|  | **336** | 0.352 | 0.432 | 0.351 | 0.37 | 0.352 |
|  | **720** | 0.413 | 0.487 | 0.409 | 0.416 | 0.383 |
|  | **Avg** | 0.342 | 0.418 | 0.343 | 0.352 | 0.329 |
| ETTm2 | **96** | 0.167 | 0.208 | 0.162 | 0.166 | 0.161 |
|  | **192** | 0.219 | 0.263 | 0.224 | 0.223 | 0.219 |
|  | **336** | 0.269 | 0.337 | 0.277 | 0.273 | 0.271 |
|  | **720** | 0.347 | 0.429 | 0.354 | 0.363 | 0.352 |
|  | **Avg** | 0.251 | 0.309 | 0.254 | 0.256 | 0.251 |
| Weather | **96** | 0.149 | 0.191 | 0.147 | 0.152 | 0.147 |
|  | **192** | 0.193 | 0.219 | 0.191 | 0.196 | 0.189 |
|  | **336** | 0.245 | 0.287 | 0.243 | 0.247 | 0.262 |
|  | **720** | 0.314 | 0.368 | 0.317 | 0.315 | 0.304 |
|  | **Avg** | 0.225 | 0.266 | 0.225 | 0.228 | 0.226 |

Table 20: Comparisons of convergence rates.

|  | methods | 96 | 192 | 336 | 720 |
|---|---|---|---|---|---|
| ETTh1 | w parameter sharing | 83/0.352 | 70/0.385 | 53/0.411 | 35/0.440 |
|  | w/o parameter sharing | 46/0.359 | 108/0.392 | 39/0.423 | 36/0.441 |
| Exchange | w parameter sharing | 71/0.081 | 51/0.179 | 82/0.328 | 32/0.842 |
|  | w/o parameter sharing | 47/0.084 | 32/0.183 | 43/0.333 | 31/0.855 |

Table 21: Comparisons of Parameter-Saving Capacity.

|  | 1 | 3 | 12 | 24 | 36 | 48 |
|---|---|---|---|---|---|---|
| Parameter Sharing | 52,416 | 58,752 | 87,264 | 125,280 | 163,296 | 201,312 |
| No Parameter Sharing | 71,424 | 115,776 | 315,360 | 581,472 | 847,584 | 1,113,696 |

## B.9 Comparisons of Convergence Rates with and without Parameter Sharing

We compare the impact of parameter sharing on the convergence rate using the ETTh1 and Exchange datasets, recording the total number of epochs and the MSE loss under the same settings.

The experimental results are shown in the Table 20, where the values represent epochs/MSE loss. We observe that the number of epochs required with or without parameter sharing on the ETTh1 dataset varies depending on the prediction length. However, for the Exchange dataset, the convergence rate is faster without parameter sharing.

Additionally, in terms of MSE loss, using parameter sharing leads to greater reductions in loss and also results in fewer parameters. Therefore, there exists a trade-off between convergence rates, loss reduction, and parameter efficiency.

## B.10 Testing of Parameter-Saving Capacity for Pre-Trained Models

For further validating the framework's parameter-saving capacity, we compared the parameter count of the PSformer under Parameter Sharing and No Parameter Sharing scenarios, including the comparison for 1-layer and 3-layer Encoders, as well as for different layers same as GPT2 models (GPT2-small, GPT2-medium, GPT2-large, and GPT2-xl) at 12-layer, 24-layer, 36-layer, and 48-layer configurations. The results are reported in the Table 21. In addition, if the hidden layer dimension is expanded from 32 to 1024 (as in GPT2), or if multi-head attention is adopted, the total number of parameters will also increase significantly.

## B.11 Discuss about Parameter Search Space

The main hyper-parameters of PSformer include: 1. the number of encoders, 2. the number of segments, and 3. the SAM hyperparameter rho. For the number of encoders, we primarily searched within 1 to 3 layers. For the number of segments, we maintain the same as the number of patches in PatchTST, we also analyzed values that divide the input length evenly (specifically: 2, 4, 8, 16, 32, 64, 128, 256), and ultimately set all prediction tasks to 32 to avoid performance improvements caused by complex hyperparameter tuning. For the SAM hyperparameter rho, we referred to SAMformer and performed a search across 11 parameter points evenly spaced in the range 0 to 1. The number of encoders and segments can be found in the ablation study in section 4.2. Additionally, for the learning rate, we mainly tested values of $1e-3$ and $1e-4$, and for the learning rate scheduler, we tested OneCycle and MultiStepLR.

## C   Limitation

Although we have discussed various aspects of PSformer, several limitations still exist:

1. SSA captures local spatiotemporal information, which makes the length of the attention matrix dependent on the channel length and local temporal length. Effectively balancing this trade-off between maintaining spatiotemporal feature extraction and reducing the attention matrix's computational overhead remains an important research direction.

2. This work lacks a theoretical explanation for the PSformer architecture's enhancement of the model's generalization ability. In the future, we will further explore possible theoretical explanations for the model's generalization capability.

3. The datasets used in this work do not yet cover all time-series datasets, such as the multivariate datasets in the TFB dataset Qiu et al. (2024) and the univariate datasets in UCR Dau et al. (2019). In the future, we will explore using a broader range of datasets to further enhance PSformer.

4. With the rapid advancement in time-series forecasting, newly proposed models may demonstrate superior predictive performance, which implies that our approach might not consistently outperform these emerging solutions.

