# OpenReview forum: "PSformer: Parameter-efficient Transformer with Segment Shared Attention for Time Series Forecasting"
_TMLR — Withdrawn by Authors_

### Review · Reviewer_k6Ko · 2025-04-30

**Summary Of Contributions:**

The paper introduces a transformer variant that (i) shares one three-layer fully-connected “PS Block” across every attention sub-module and fusion stage, and (ii) replaces patch-wise self-attention with a two-stage Segment-Shared Attention (SSA) that works on local spatio-temporal “segments.” Together, these two ideas yield a parameter-efficient yet expressive backbone for multivariate time-series forecasting.

**Audience:**

Yes

**Broader Impact Concerns:**

N.A.

**Claims And Evidence:**

Yes

**Requested Changes:**

1. Elaborate on novelty relative to Crossformer, Reformer weight sharing, and ALBERT in Related Work; include a concise comparison table or paragraph.

2. Release runnable training scripts that reproduce the main table with a single command.

**Strengths And Weaknesses:**

Pros:
1. Elegant combination of parameter sharing and segment-wise attention for time series.

2. Clear algorithmic description; implementation details (RevIN, SAM) support reproducibility.

3. Broad benchmark coverage and thorough ablation studies.

4. Impressive parameter and runtime efficiency—tens of thousands of parameters instead of millions.

5. Well-structured paper with a detailed appendix.

Cons:

1. Novelty may appear incremental: SSA is reminiscent of Crossformer-style cross-dimension attention and ALBERT-style weight sharing.

2. Lacks a theoretical explanation for why the two SSA stages or PS sharing improve generalisation.

3. No statistical significance tests; all results are single-seed.

4. Missing comparisons against size-matched tiny versions of PatchTST/SAMformer—unclear how much gain comes purely from being small.

5. Minor writing issues: occasional grammar slips and duplicated background paragraphs.

---

> ### Author Response · Authors · 2025-06-02
>
> Thank you for your valuable comments, we address each of your questions in the following response:
>
> **1.Elaborate on novelty relative to Crossformer, Reformer weight  sharing, and ALBERT in Related Work; include a concise comparison table  or paragraph.**
>
> PSformer introduces key innovations beyond existing parameter sharing and cross-channel attention approaches: (1) Our Segment Shared Attention (SSA) uniquely processes local spatio-temporal segments, jointly capturing cross-channel and temporal relationships - unlike SAMformer (cross-channel only) or Crossformer (separate routing of dimensions); (2) The parameter sharing scheme radically shares all parameters within each PSEncoder (across SSA modules and fusion stages) while maintaining independence across encoders, differing from ALBERT-style cross-layer sharing. Experiments confirm this achieves both state-of-the-art performance and parameter efficiency. We have revised Section 2.2, 2.3 according to your suggestions and included a concise comparison in Table 1.
>
> **2.Lacks a theoretical explanation for why the two SSA stages or PS sharing improve generalisation.**
>
> The SSA stage captures local spatio-temporal information while reducing parameter count through parameter sharing. Compared to non-parameter-shared approaches, this maintains the model's ability to process spatio-temporal information while potentially lowering the risk of overfitting due to fewer parameters. As shown in Figure 3, the validation loss decreases and remains stable at a low level during training.
>
> However, providing a rigorous theoretical explanation for this phenomenon may be challenging. We have documented this in the limitations section. Empirically, more complex model architectures (with higher parameter counts, more layers, etc.) tend to exhibit reduced robustness and often require additional techniques to improve generalization, such as Dropout, LayerNorm,  or L1 regularization.
>
> **3.Missing comparisons against size-matched tiny versions of  PatchTST/SAMformer—unclear how much gain comes purely from being small.**
>
> The total number of parameters in PSformer is comparable to that of SAMformer and RLinear, as shown in Table 8. Therefore, when compared at a size-matched level, PSformer performs better than SAMformer.
>
> **4.Minor writing issues: occasional grammar slips and duplicated background paragraphs.**
>
> Thank you for kindly pointing out the writing issues. We've fixed potential spelling errors in the revised version.
>
> **5.Release runnable training scripts that reproduce the main table with a single command.**
>
> We've updated the code, which can simply run all_script_once_run.sh to generate the main table results, saved as CSV file in the root directory. Besides, we evaluated the statistical measures across multiple runs with different random seeds in six datasets (excluding traffic and electricity for time reasons), as documented in Table 14 of the paper.
>
> Finally, we sincerely thank you for your review feedback and the time you have dedicated.

---

### Review · Reviewer_ADoj · 2025-05-11

**Summary Of Contributions:**

The paper studies an important problem of time series forecasting. Generally, the paper is well-written and easy to follow. The authors propose a parameter-efficient transformer with segment shared attention. Extensive experiments show the effectiveness of the proposed method.

**Audience:**

Yes

**Broader Impact Concerns:**

Please see the contributions.

**Claims And Evidence:**

Yes

**Requested Changes:**

Please see the weaknesses.

**Strengths And Weaknesses:**

Strengths:
1. The paper is well-written and easy to follow.
2. The proposed PSformer is innovative and sound, including a parameter sharing module (PS) and Segment Shared Attention (SSA).
3. Extensive experiments show the effectiveness of the proposed method.

Weaknesses:
1. Existing studies often conduct experiments on several classic datasets, such as traffic, electricity, weather, ECL, and ILI. Please refer to TimeLLM and TimeCMA.

[1]. Time-LLM: Time Series Forecasting by Reprogramming Large Language Models, ICLR 2024.

[2]. TimeCMA: Towards LLM-Empowered Multivariate Time Series Forecasting via Cross-Modality Alignment, AAAI 2025.

It is suggested to conduct more experiments on the full datasets to comprehensively demonstrate the effectiveness of the proposed method.

2. More recent baselines are required as follows.

[3]. DUET: Dual Clustering Enhanced Multivariate Time Series Forecasting, KDD 2025.

[4]. UniTS: A Unified Multi-Task Time Series Model, NeurIPS 2024.

3. It would be more interesting to discuss the advantages and disadvantages of the proposed method. It is encouraged to show the limitations.

---

> ### Author Response · Authors · 2025-06-02
>
> **Add datasets and benchmarks for comparison, and showcase limitations**
>
> Thank you for your valuable feedback. Upon reviewing the datasets employed in TimeCMA and TimeLLM, we noted their inclusion of the ILI and FRED-MD datasets. In response to this observation, we conducted additional performance evaluations of PSformer on these datasets, supplemented by comparative analyses against TimeCMA and relevant benchmark models. Detailed results are documented in Appendix B.8.1 and Table 17, where PSformer demonstrates competitive performance on the ILI dataset, while TimeCMA achieves superior results on FRED-MD.
>
> This suggests that model performance may vary across different datasets. Conducting broader comparisons on more extensive datasets would provide a more comprehensive understanding of each model's strengths and weaknesses—a gap we identify as a limitation and intend to investigate in future research, so we document it in Limitations section.
>
> Additionally, another limitation of PSformer lies in its increased attention matrix size when capturing spatio-temporal dependencies. Further research is needed to reduce the attention matrix size through methods like sparse attention. Please refer to Appendix C of our revised manuscript for detailed discussions.
>
> Finally, we sincerely thank you for your review feedback and the time you have dedicated.

---

### Review · Reviewer_1ax9 · 2025-05-21

**Summary Of Contributions:**

The paper proposes PSformer, a parameter-efficient Transformer for multivariate time series forecasting. It introduces two key components: a Parameter Sharing Block (PS Block) to reduce model size and a Segment Shared Attention (SSA) mechanism to capture temporal and cross-variable dependencies jointly. It performs state-of-the-art on 7 out of 8 benchmark datasets with significantly fewer parameters, demonstrating strong accuracy, scalability, and generalization.

**Audience:**

Yes

**Claims And Evidence:**

Yes

**Requested Changes:**

The submission presents a well-motivated and technically sound approach to multivariate time series forecasting. The methodology is clearly explained, and the experiments demonstrate strong empirical results that outperform several baselines.  Thus, I recommend acceptance as is.

**Strengths And Weaknesses:**

**Strengths**

1. The paper is clearly written and easy to understand.
2. It proposes a novel Transformer model, PSformer, which combines parameter sharing with segment-based attention in a well-motivated way.
3. The model shows strong performance, achieving state-of-the-art results on 7 out of 8 benchmark datasets, surpassing both lightweight (e.g., RLinear and TSMixer) and large pre-trained (e.g., MOMENT and GPT4TS) models.
4. The ablation studies are thorough and well-designed, providing solid evidence for the effectiveness of each proposed component.

---

**Weaknesses**

The overall novelty of the paper is limited. PSformer integrates several known enhancements from recent time series forecasting work, but does not provide substantial new theoretical or architectural insights. However, considering the acceptance criteria for TMLR that requires claims made in the submission to be supported by accurate, convincing, and clear evidence, I am inclined to accept this work.

---

> ### Author Response · Authors · 2025-06-02
>
> Thank you for your valuable review comments. The structure of PSformer is concise and efficient. Although the ideas of parameter sharing and cross-channel attention are explored in some works, there are still significant differences in structural design, our work differs from previous studies in the following aspects:
>
> First, Segment Shared Attention directly applies the attention mechanism to local spatio-temporal segments, enhancing the ability to process cross-channel local temporal information. While for other cross-channel designs，the SAMformer applies attention to the cross-channel dimension without simultaneously considering temporal dimension information. Crossformer uses a routing mechanism to separately characterize information in the temporal and spatial dimensions, then fuses different information through a decoder. In contrast, PSformer does not adopt the scheme of routing mechanisms for single dimensions followed by multi-dimensional information fusion. Instead, it directly applies attention to local spatio-temporal information, reducing model complexity. It uses a parameter-sharing mechanism to reduce model parameters, and this parameter-sharing scheme is quite radical: the parameters of one PSBlock are shared between the two SSA modules and the fusion stage within the PSEncoder, which is significantly different from past parameter-sharing designs.
>
> In terms of specific comparisons of parameter-sharing designs，ALBERT uses cross-layer parameter sharing but does not share parameters within each layer. PSformer does not share parameters across PSEncoders but shares parameters between the SSA and Fusion stage within an Encoder. We have added relevant comparisons and summaries in Section 2.3 of the revised paper.
> Besides, since SSA applies attention to local spatio-temporal segments and Q, K, V are derived from the shared PSBlock, we can study the attention relationships between channels by correlating attention with specific time series. Additionally, the combination of SSA and the parameter-sharing experiments show competitive performance and parameter-efficient.
>
> Finally, we sincerely thank you for your review feedback and the time you have dedicated.

---

### Note · Authors · 2025-07-28

I have read and agree with the venue's withdrawal policy on behalf of myself and my co-authors.